# Fecal Microbiota Transplantation from Mice Receiving Magnetic Mitohormesis Treatment Reverses High-Fat Diet-Induced Metabolic and Osteogenic Dysfunction

**DOI:** 10.3390/ijms26125450

**Published:** 2025-06-06

**Authors:** Jun Kit Craig Wong, Bharati Kadamb Patel, Yee Kit Tai, Tuan Zea Tan, Wei Wei Thwe Khine, Way Cherng Chen, Marek Kukumberg, Jianhong Ching, Lye Siang Lee, Kee Voon Chua, Tsze Yin Tan, Kwan Yu Wu, Xizhe Bai, Jan Nikolas Iversen, Kristy Purnamawati, Rufaihah Abdul Jalil, Alan Prem Kumar, Yuan Kun Lee, Shabbir M. Moochhala, Alfredo Franco-Obregón

**Affiliations:** 1Department of Surgery, Yong Loo Lin School of Medicine, National University of Singapore, Singapore 119228, Singapore; surwjkc@nus.edu.sg (J.K.C.W.); surbkp@nus.edu.sg (B.K.P.); lesleywu@nus.edu.sg (K.Y.W.); xizhebai@gmail.com (X.B.); nikolas.iversen@u.nus.edu (J.N.I.); prnkristy@gmail.com (K.P.); rufaihah_abdul_jalil@tp.edu.sg (R.A.J.); phcsmm@nus.edu.sg (S.M.M.); 2Institute of Health Technology and Innovation (iHealthtech), National University of Singapore, Singapore 117599, Singapore; 3BICEPS Lab (Biolonic Currents Electromagnetic Pulsing Systems), National University of Singapore, Singapore 117599, Singapore; 4NUS Centre for Cancer Research (N2CR), Yong Loo Lin School of Medicine, National University of Singapore, Singapore 117597, Singapore; apkumar@nus.edu.sg; 5Genomics and Data Analytics Core, Cancer Science Institute of Singapore, National University of Singapore, Singapore 117599, Singapore; csittz@nus.edu.sg; 6Department of Microbiology and Immunology, Yong Loo Lin School of Medicine, National University of Singapore, Singapore 117545, Singapore; weithwe@gmail.com (W.W.T.K.); micleeyk@nus.edu.sg (Y.K.L.); 7Bruker Singapore Pte Ltd., 30 Biopolis St., Singapore 138671, Singapore; way_cherng.chen@bruker.com; 8Healthy Longevity Translational Research Programme, Yong Loo Lin School of Medicine, National University of Singapore, Singapore 119228, Singapore; marek.kukumberg@nus.edu.sg; 9Stem Cell Core Facility, Healthy Longevity Translational Research Programme, Yong Loo Lin School of Medicine, National University of Singapore, Singapore 117544, Singapore; 10Cardiovascular and Metabolic Disorders Programme, Duke-NUS Medical School, Singapore 169857, Singapore; jianhong.ching@duke-nus.edu.sg (J.C.); lyesiang.lee@duke-nus.edu.sg (L.S.L.); pris.ckv@duke-nus.edu.sg (K.V.C.); zing.tan@duke-nus.edu.sg (T.Y.T.); 11KK Research Centre, KK Women’s and Children’s Hospital, Singapore 229899, Singapore; 12Future Foods Domain, School of Applied Science, Temasek Polytechnic, Singapore 529757, Singapore; 13Department of Pharmacology, Yong Loo Lin School of Medicine, National University of Singapore, Singapore 117600, Singapore; 14Department of Physiology, Yong Loo Lin School of Medicine, National University of Singapore, Singapore 117593, Singapore; 15Competence Center for Applied Biotechnology and Molecular Medicine, University of Zürich, 8057 Zürich, Switzerland

**Keywords:** gut microbiome, PEMF therapy, non-alcoholic fatty liver disease (NAFLD), non-alcoholic steatohepatitis (NASH), obesity, ceramides, osteogenesis

## Abstract

This study compared the metabolic consequences of fecal microbiota transplantation (FMT) from donor mice that had been either administered pulsed electromagnetic field (PEMF) therapy or exercised to recipient mice fed a high-fat diet (HFD). Eight weeks of PEMF treatment (10 min/week) enhanced PGC-1α-associated mitochondrial and metabolic gene expression in white and brown adipose to a greater degree than eight weeks of exercise (30–40 min/week). FMT from PEMF-treated donor mice recapitulated these adipogenic adaptations in HFD-fed recipient mice more faithfully than FMT from exercised donors. Direct PEMF treatment altered hepatic phospholipid composition, reducing long-chain ceramides (C16:0) and increasing very long-chain ceramides (C24:0), which could be transferred to PEMF-FMT recipient mice. FMT from PEMF-treated mice was also more effective at recovering glucose tolerance than FMT from exercised mice. PEMF treatment also enhanced bone density in both donor and HFD recipient mice. The gut Firmicutes/Bacteroidetes (F/B) ratio was lowest in both the directly PEMF-exposed and PEMF-FMT recipient mouse groups, consistent with a leaner phenotype. PEMF treatment, either directly applied or via FMT, enhanced adipose thermogenesis, ceramide levels, bone density, hepatic lipids, F/B ratio, and inflammatory blood biomarkers more than exercise. PEMF therapy may represent a non-invasive and non-strenuous method to ameliorate metabolic disorders.

## 1. Introduction

Exercise stimulates the metabolic remodeling of skeletal muscle and adipose tissues. This adaptive response is the consequence of a system of paracrine crosstalk between muscle, adipose, hepatic, and bone tissues, in descending order of impact [1]. Muscle-induced systemic crosstalk ultimately enhances the diversity and functional capacity of the gut microbiota [2]. Epidemiological and omics-based studies have underscored the importance of the gut microbiome in shaping human health and susceptibility to disease [3], by attenuating chronic systemic inflammation and lipotoxicity that, in turn, promotes metabolic balance, improves immunity, and supports bone and muscle maintenance, thereby offsetting the onset of disorders such as diabetes and liver disease [4,5]. Exercise has emerged as a key modulator of gut microbial diversity and function [6], alongside genetics, antibiotics, and diet [7]. On the one hand, exercise stimulates intestinal motility, accelerating the transit of food through the gastrointestinal tract, which influences the composition of the gut microbiota [8]. On the other hand, a form of exercise-induced muscle–gut crosstalk exists, mediated by lactate and short-chain fatty acids (SCFAs), respectively, that likewise shapes the composition of the comprehensive gut microbiota [9,10].

The gut microbiota is principally comprised of two phyla, *Firmicutes* and *Bacteroidetes* [11]. The *Firmicutes* phylum consists of over 200 genera, including *Lactobacillus*, *Bacillus*, *Clostridium*, *Enterococcus*, and *Ruminococcus*, that predominantly reside in the upper regions of the small intestine, are largely concerned with extracting energy from foods, and contribute to energy storage as adipose tissue. The *Bacteroidetes*, which include genera such as *Prevotella* and *Bacteroides*, reside in the colon and are largely responsible for producing the SCFAs from insoluble fibers [12]. Moderate exercise reduces the *Firmicutes*-to-*Bacteroidetes* (F/B) ratio [6,13,14,15], whereas high-fat diets (HFDs) increase the F/B ratio [16]. Accordingly, a lower F/B ratio is associated with lower adiposity and an improved metabolic profile in humans and animals [17,18].

Obesity-related microbiome disruption undermines bone maintenance. HFD-associated dysbiosis [19] promotes intestinal inflammation and weakens the intestinal mucosal barrier, ultimately leading to systemic inflammation and impaired bone metabolism [20]. HFDs further exacerbate inflammation by reducing omega-3 long-chain polyunsaturated fatty acids that protect against bone loss [21]. Additionally, HFDs stimulate intestinal serotonin production, exacerbating inflammation-mediated bone loss [20]. Osteoporotic patients hence exhibit distinctive gut microbiota profiles [5] that are associated with compromised bone metabolism [22], bone mineral resorption, and immune dysregulation [23]. Osteoporosis, obesity, and gut microbiome dysbiosis are hence humorally intertwined.

Given the global rise in obesity and associated metabolic disorders, accompanied by an increasingly sedentary population, an urgent need exists for the development of effective exercise mimetics that are less burdensome than traditional exercise programs. One potential alternative is electromagnetic field therapy [24]. Brief (10 min) exposure to extremely low-frequency (Hz–100 Hz) and low-amplitude (1.5 milliTesla; mT) pulsed electromagnetic fields (PEMFs) have been shown to stimulate in vitro (single exposures) and in vivo (weekly exposures) myogenesis along the oxidative phenotype, recapitulating the key metabolic hallmarks of exercise in cultured muscle cells [25,26,27], mice [15,26], and humans [28,29]. A key feature of the oxidative muscle phenotype is an enrichment of the mitochondria mediated by the transcriptional co-activator, PPAR-γ co-activator-1 α (PGC1-α) [30], that is upregulated by both exercise [30,31] and PEMF exposure [15,25]. Exercise- [31] and PEMF-stimulated [15] PGC1-α expression is translated to adipose tissues via the anti-inflammatory nature of the muscle secretome, a ramification of muscle–adipose crosstalk [31,32]. Systemic adaptations are fundamentally manifested at the level of the mitochondrion [24].

Calcium-signaling cascades are implicated in the biological responses to electromagnetic exposure [33,34,35,36,37]. The calcium-permeable Transient Receptor Potential Cation Canonical Channel 1 (TRPC1) has recently emerged as a key component of a Ca^2+^-mitochondrial signaling axis invoked during magnetoreception [36,38]. Oxidative muscle development and function require the participation of TRPC1 [39,40] and mitochondria [41]. Oxidative muscles are recruited during fatiguing tasks and help postural maintenance against the constant force of gravity, making them fundamental for establishing basal metabolism [24]. Underscoring the mutual activation of TRPC1 by exercise [42] and magnetic exposure [15,25], both forms of biophysical stimulation promote oxidative muscle development [15,25] and reduce the F/B ratio [15,25] characteristic of a leaner phenotype [43].

Fecal microbiota transplantation (FMT) involves the transfer of fecal matter from donors to recipients with the aim of altering the composition of the recipient’s gut microorganisms to match that of the donor [44]. FMT-mediated microbiome species transfer allows for the potential transmittance of microbiome-governed metabolic characteristics from donors to recipients [45]. Accordingly, FMT has been demonstrated capable of modulating the inflammatory, immune, and metabolic statuses of human recipients [46], making it a promising therapeutic intervention for a range of diseases [47], including osteoporosis [48] and muscular disorders [49].

This study investigated the consequences of 8 weeks of PEMF treatment or exercise training on mouse adipose, bone, muscle, and hepatic metabolic adaptations and whether they could be recapitulated in obese mice by FMT. The findings indicate that magnetic therapy and exercise share systemic adaptations, some of which could be conferred by the gut microbiome.

## 2. Results

### 2.1. Study Workflow and Weight Assessment

Two mouse experimental paradigms were investigated. In paradigm 1 (Figure 1a), healthy C57/BL6J mice were designated as the FMT donors. These mice were subjected to one of the four interventions for 8 weeks: (1) sham exposure (−E/−P); (2) 1.5 mT PEMF once a week (-E/+P); (3) running exercise twice a week (+E/−P); or (4) a combination of both (+E/+P) (Figure 1a). All groups exhibited consistent weight gain over the 8-week period, with no significant differences between the groups for weight and food intake (Figure 1(bi,bii)). Dry fecal matter was collected twice a week from mice exposed to PEMF (−E/+P) or exercise (+E/−P) up to week 8 and stored for later use in paradigm 2.

In paradigm 2 (Figure 1c), C57/BL6J mice were fed a high-fat diet (HFD) for 8 weeks and were designated as fecal microbiota transplantation (FMT)-recipient mice. Four weeks before the FMT initiation via oral gavage, the native gut microbiota of these HFD mice was eradicated using antibiotics (Figure 1c). After 8 weeks of the HFD, the mice exhibited a ~1.4-fold increase in body weight compared to their starting weight (Figure 1(di)). Their diet was then switched to standard chow, whereupon they received either biweekly PEMF-FMT, Exe-FMT (exercise-derived FMT), or Sham FMT (saline control). A separate cohort remained on an HFD for an additional 8 weeks (16-week HFD, yellow line), resulting in a 1.8-fold increase in body weight relative to baseline. In contrast, the mice receiving FMT while maintained on standard chow exhibited stabilized body weight following the intervention (Figure 1(dii)). By week 16, the PEMF-FMT cohort (blue line) demonstrated the greatest relative reduction in body weight (1.36-fold), followed by the Exe-FMT (green line; 1.42-fold) and Sham FMT (red line; 1.48-fold) cohorts. No differences in food intake were observed amongst the groups (Figure 1(diii)).

As a matter of general convention, red (in the symbols, bars, and boxes) refers to the control or reference condition. For the FMT donor mice, red corresponds to the unexposed or nonexercised mice scenario, and for the FMT recipient mice, red corresponds to the sham saline gavage. Blue (in the symbols, bars, and boxes) refers to the PEMF exposure scenario. For the FMT donor mice, blue corresponds to the mice exposed to PEMF, and for the FMT recipient mice, blue corresponds to the mice receiving FMT originating from the PEMF-exposed mice. Green (in the symbols, bars, and boxes) refers to the exercised scenario. For the FMT donor mice, green corresponds to the mice having been exercised, and for the FMT recipient mice, green corresponds to the mice receiving FMT originating from the exercised mice.

### 2.2. Modification of Metabolism-Regulating Gene Expression in White and Brown Adipose Tissues

Exercise [6,13,14,15] and an HFD [50] exert disparate effects over the *Firmicutes*-to-*Bacteroidetes* (F/B) ratio of the gut microbiome, reducing and raising the ratio, respectively. PEMF exposure was also shown to reduce the (F/B) ratio in conjunction with preferentially enhancing the metabolic capacity of white adipose tissue (WAT) over brown adipose tissue (BAT) [15,26]. To more closely examine this gut–adipose interaction, changes in gene expression in the WAT and BAT from FMT donor and FMT recipient HFD mice, in the context of PEMF exposure and exercise, were analyzed using real-time quantitative reverse transcription PCR (qRT-PCR). With reference to the WAT, direct PEMF exposure (−E/+P) significantly upregulated the expression of *Pgc1a*, a key gene involved in mitochondrial biogenesis (Figure 2(ai)). A similar upregulation was observed in PEMF-FMT HFD-recipient mice (Figure 2(aii)), whereas neither exercise (+E/−P) nor the Exe-FMT induced a statistically significant upregulation of *Pgc1a*. Cytochrome c oxidase subunit 7A1 (*Cox7a1*), a component of the mitochondrial respiratory chain, was upregulated by direct PEMF exposure (−E/+P) (Figure 2(bi)) but was downregulated in the WAT of PEMF-FMT recipients (Figure 2(bii)). This result agrees with a published study showing that 6 weeks of an HFD downregulated *Pgc1a* and upregulated *Cox7a1* [51], whereas here it is shown that PEMF-FMT upregulated *Pgc1a* (Figure 2(aii)) and downregulated *Cox7a1* (Figure 2(bii)). The PEMF intervention, whether applied directly or mediated through microbiome transfer, exerted significant mitochondrial effects.

PEMF exposure (−E/+P) significantly increased the expression of genes related to glucose transport (*Glut4*; Figure 2(ci)) and protein synthesis (*Rpl23*; Figure 2(di)), whereas PEMF-FMT did not produce significant changes in these genes (Figure 2(cii,dii)). By contrast, exercise (+E/−P) did not elevate *Glut4* or *Rpl23* significantly, although Exe-FMT increased *Glut4* expression relative to HFD controls (Figure 2(cii)). The thermogenic genes, *Cebpa* and *Prdm16*, were both upregulated by PEMF exposure (−E/+P) (Figure 2(ei,fi)), whereas only *Cebpa* was increased by exercise (+E/−P) (Figure 2(ei)). In all the FMT cohorts, the increases in *Cebpa* and *Prdm16* expression relative to the HFD controls did not reach statistical significance (Figure 2(eii,fii)).

NAMPT, which catalyzes the rate-limiting step in NAD^+^ biosynthesis and is a critical co-factor for mitochondrial respiration [52], and UCP1, associated with adaptive non-shivering thermogenesis during the beiging of white adipose tissue [31], were reciprocally modulated by the combination of PEMF exposure and exercise (+E/+P). Specifically, +E/+P reduced *Nampt* expression (Figure 2(gi)) while increasing *Ucp1* expression (Figure 2(hi)) compared to the controls (−E/−P). The combination of PEMF exposure and exercise (+E/+P) was previously shown to maximally induce *Ucp1* expression in WAT [15,26]. By contrast, FMT administration did not significantly alter *Nampt* or *Ucp1* expression (Figure 2(gii,hii)), indicating that adipose thermogenesis was not influenced by FMT.

In the BAT of the donor mice, direct PEMF exposure (−E/+P) and exercise (+E/−P) significantly upregulated the expressions of *Pgc1a*, *Cox7a1*, *Glut4*, *and Rpl23* (Figure 3(a–di)) relative to the control mice (−E/−P), whereas *Cebpa*, *Prdm16*, and *Nampt* remained unchanged (Figure 3(e,fi)). Notably, *Ucp1* gene expression was downregulated by both PEMF exposure (−E/+P) and exercise (+E/−P) (Figure 3(hi)). In contrast to the WAT, the BAT from the PEMF-FMT and Exe-FMT groups showed consistent upregulations of *Pgc1a*, *Cox7a1*, *Glut4*, and *Rpl23* relative to Sham FMT, without altering the expressions of *Cebpa*, *Prdm16*, *Nampt*, and *Ucp1* (Figure 3(a–hii)).

### 2.3. Modulation of the Oxidative Character of Muscle

The soleus and extensor digitorum longus (EDL) muscles represent muscles of opposing metabolic capabilities, characterized by high and low oxidative capacities, respectively [53]. The *PGC-1α* expression, oxidative character, and antioxidant capacity of the soleus muscles of mice were previously shown to be enhanced by an analogous PEMF exposure and exercise paradigm [15]. In this present study, the HFD-fed recipient mice that received Exe-FMT significantly increased solei *PGC-1α* and *Ppar-α* expression along with other PGC-1α-related genes, such as *Nrf2*, *Tfeb*, and *Sirt1* (Figure 4(ai–av)). PEMF-FMT produced the greatest increase in *Nrf2* expression in the solei, with *Pgc1a*, *Pparα*, *Tfeb*, and *Sirt1* also trending to be higher than in Sham FMT. At the protein level, PEMF-FMT increased PGC-1α and Type IIA protein expression in EDL muscles (Figure 4(bi,bii)), with minimal changes in Type IIB expression across the FMT groups (Figure 4(biii)). A higher type IIA/IIB ratio reflects a greater oxidative phenotype and was most elevated in the PEMF-FMT group (Figure 4(biv)). Both Exe-FMT and PEMF-FMT enhanced the oxidative character of muscle in the HFD-recipient mice.

**Figure 3 ijms-26-05450-f003:**
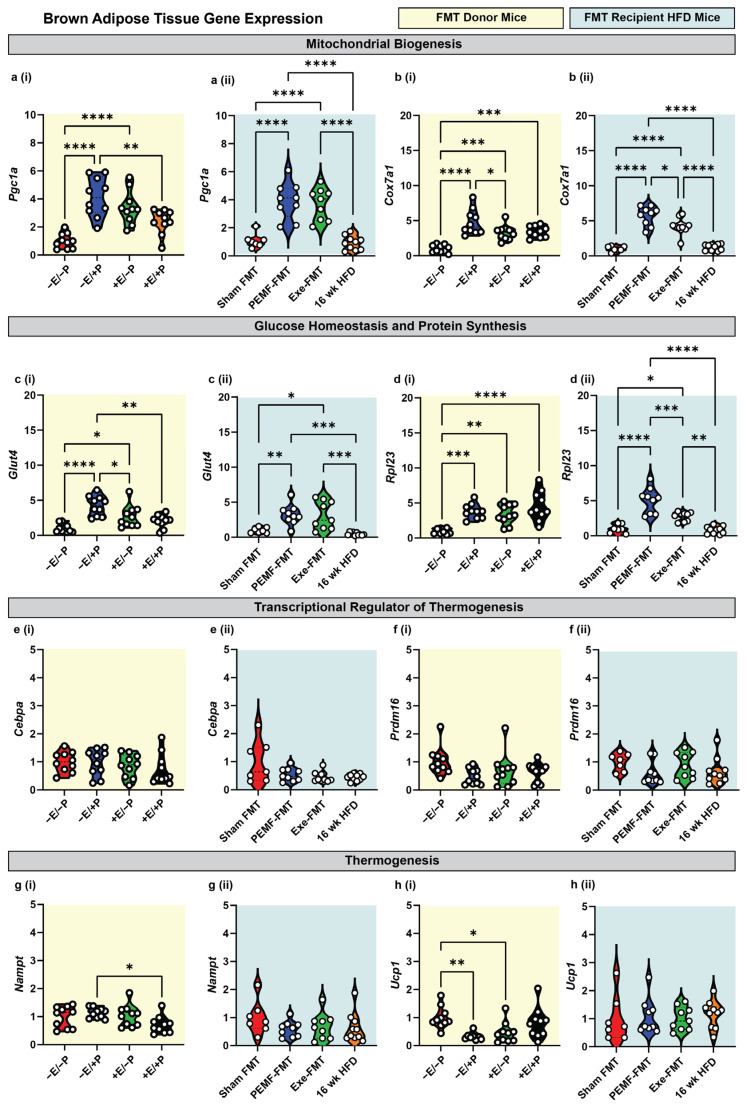
Modulation of metabolic genes in brown adipose tissue of donor and FMT recipient HFD mice. Gene expression analysis 2^(−ΔΔCT)^ of mitochondrial biogenesis markers: (**a**) *Pgc1a* and (**b**) *Cox7a1*; glucose homeostasis/protein synthesis markers: (**c**) *Glut4* and (**d**) *Rpl23*; transcriptional regulators of thermogenesis: (**e**) *Cebpa* and (**f**) *Prdm16*; and thermogenesis effectors: (**g**) *Nampt* and (**h**) *Ucp1.* Yellow backgrounds (**i**) indicate FMT donors (paradigm 1; E = exercise; P = PEMF) and light blue backgrounds (**ii**) indicate FMT recipients (paradigm 2; Exe = exercise). Each white dot within the violin plots represents an individual animal. Statistical significance was determined by a one-way ANOVA with Sidak’s post hoc test (* *p* < 0.05, ** *p* < 0.01, *** *p* < 0.001, and **** *p* < 0.0001).

### 2.4. Changes in Bone Density

Metabolic resilience influences bone health [54]. PEMF exposure (−E/+P) significantly increased median tibia bone volume by 45% (~21 mm^3^ vs. ~15 mm^3^ in control mice) (Figure 4c) measured by micro-computed tomography (micro-CT). Exercise alone (+E/−P; ~16 mm^3^), or in combination with PEMFs (+E/+P; ~15 mm^3^), did not significantly alter the bone volume. Bone volume, encompassing both cortical and trabecular bone contributions, reflects overall bone density and structure (Figure 4d). PEMF exposure produced cortical bone thickening and increased trabeculae bone density (Figure 4e; yellow arrows), indicating general improvements in the bone volume. By contrast, exercise alone, or combined exercise and PEMF (+E/+P), had a lesser impact on cortical and trabecular bone remodeling.

### 2.5. Specific Bone Indices

PEMF-exposed FMT donor mice (−E/+P) exhibited a significantly enhanced cortical bone structure as reflected by increases in median cortical thickness (+9%), cortical volume (+15%), and cortical bone mineral density (+1.1%) compared to the control mice (−E/−P) (Figure 5(a–ci)). The Sham FMT mice, which represent mice previously fed with 8 weeks of an HFD and given Sham FMT, showed a significant reduction in cortical BMD compared to the control donor mice (Figure 5(ciii)). Notably, the HFD recipient mice that had received PEMF-FMT showed improvements in bone cortical volume (+7.7%) and mineral density (+1.2%) relative to the mice on a 16-week HFD (Figure 5(b–cii)). Both the PEMF-FMT and Exe-FMT recipient mice exhibited higher cortical bone mineral density (+1.2% and +1.8%, respectively) than the mice on the 16-week HFD (Figure 5(cii)). However, Exe-FMT or PEMF-FMT did not significantly affect cortical thickness (Figure 5(aii)). PEMF exposure also enhanced trabecular bone development, increasing trabecular thickness (+13.8%) and number (+19.3%) and reducing trabecular separation, compared to the control mice (−3%) (Figure 5(d–fi)). Similarly, PEMF-FMT recipient mice also showed improvements in trabecular thickness (+17.5%), trabecular number (+15.3%), and trabecular percentage (+34.2%) compared to the 16-week HFD mice (Figure 5(d–fii)). Overall, PEMF exposure and PEMF-FMT showed the most consistent and significant improvement in bone health indices compared to the control or Sham FMT-fed mice.

**Figure 5 ijms-26-05450-f005:**
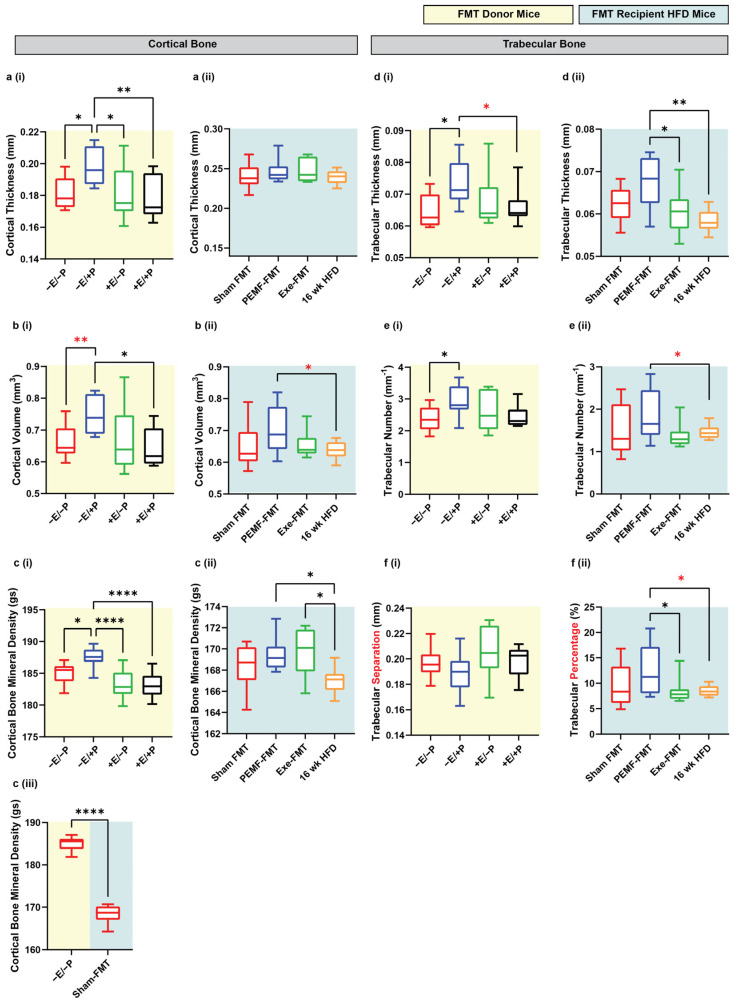
Changes in bone indices in FMT donor and recipient HFD mice. Cortical and trabecular tibial indices were analyzed in (**i**) FMT donor mice (yellow background) and (**ii**) FMT recipient HFD mice (light blue background). Parameters included the following: (**a**) cortical thickness (mm), (**b**) cortical volume (mm^3^), (**c**) cortical BMD (g/cm^3^), (**d**) trabecular thickness (mm), (**e**) trabecular number (/mm), and (**f**) either (**i**) trabecular separation (mm) for donors or (**ii**) trabecular percentage (%) for recipients. (**c**) (**iii**) Comparison between donor (−E/−P) and Sham FMT recipient mice from panels (**c**) (**i**,**ii**), highlighting differences in cortical BMD. Data represents n = 8–10 mice per group. All analyses were performed using a one-way ANOVA with Sidak’s multiple comparisons test (* *p* < 0.05, ** *p* < 0.01, and **** *p* < 0.0001). Red asterisks denote significant inter-group differences determined by Student’s *t*-tests (* *p* < 0.05, ** *p* < 0.01). E = exercise and P = PEMF.

### 2.6. Blood Glucose Tolerance and Metabolic Biomarkers in HFD Mice

Blood biomarkers were analyzed using ELISA and multiplex assays. Unfasted insulin levels were higher in Exe-FMT recipient mice than in the Sham FMT or PEMF-FMT groups (Figure 6a). Leptin, an adipokine associated with obesity [55], was highest in the mice fed an HFD for 16 weeks and lowest in the PEMF-FMT group (Figure 6b). Conversely, adiponectin, an adipokine-myokine that is shown to improve insulin sensitivity [55], was significantly elevated in the PEMF-FMT and Exe-FMT groups relative to the Sham FMT or HFD groups (Figure 6c). Vascular endothelial growth factor (VEGF), a well-established exerkine [56], showed the most pronounced increase in the PEMF-FMT recipient mice relative to the other groups (Figure 6d). The provision of Exe-FMT to HFD-fed mice resulted in reduced plasma IL-6 levels compared to Sham FMT (Figure 6e). Monocyte chemoattractant protein-1 (MCP-1), related to obesity and chronic inflammation [57], was significantly upregulated in the 16-week HFD mice relative to the other FMT groups (Figure 6f). PEMF-FMT appears to have produced the greatest improvements with reference to the serum levels of classical adipokines [55].

Adipokine gene expression was next examined in the white and brown adipose tissue (WAT and BAT) from the donor and recipient mice. In the FMT donor mice, exercise (+E/−P) and combined PEMF and exercise (+E/+P) reduced WAT *AdipoQ* gene expression relative to the control mice (−E/−P), whereas PEMF exposure alone (−E/+P) preserved the transcript levels (Figure 6(gi)). By contrast, the PEMF-FMT recipients exhibited a lower WAT *AdipoQ* expression than the Sham FMT or Exe-FMT recipients (Figure 6(gii)), reversing the trend observed with direct PEMF exposure. With reference to *Leptin*, the combination of PEMF treatment and exercise (+E/+P) resulted in its greatest reduction in the WAT, contrasting with its elevation in the 16-week HFD mice relative to PEMF-FMT (Figure 6(hii)). The *Paqr4* (Progestin and AdipoQ Receptor 4) gene, involved in ceramide signaling [58], exhibited a statistically significant downregulation in the direct PEMF treatment group (Figure 6i(i)), while showing a marked elevation in the mice maintained on the HFD for 16 weeks (Figure 6i(ii)). In the BAT, both direct PEMF exposure and PEMF-FMT significantly upregulated *AdipoQ* expression compared to their respective controls (Figure 6(ji,jii)). Exercise (+E/−P) downregulated *Leptin* expression relative to the control, while other groups showed no significant changes (Figure 6(ki)). Notably, *Leptin* expression remained the highest in the 16-week HFD mice (Figure 6(kii)). Similar to the WAT, the BAT *Parqr4* expression was the lowest in the direct PEMF exposure group and exhibited a similar trend to the PEMF-FMT recipients.

The ability of PEMF-FMT to modulate blood glucose control was next investigated. During the final week of the FMT administration, the FMT recipient mice were injected with a bolus of glucose, and their blood glucose levels were monitored using a glucometer at various intervals up to 2 h (Figure 7a). Compared to either the Sham FMT or 16-week HFD groups, the PEMF-FMT recipients demonstrated the best management of blood glucose levels, exhibiting the fastest return to baseline following the glucose challenge (Figure 7a,b).

**Figure 6 ijms-26-05450-f006:**
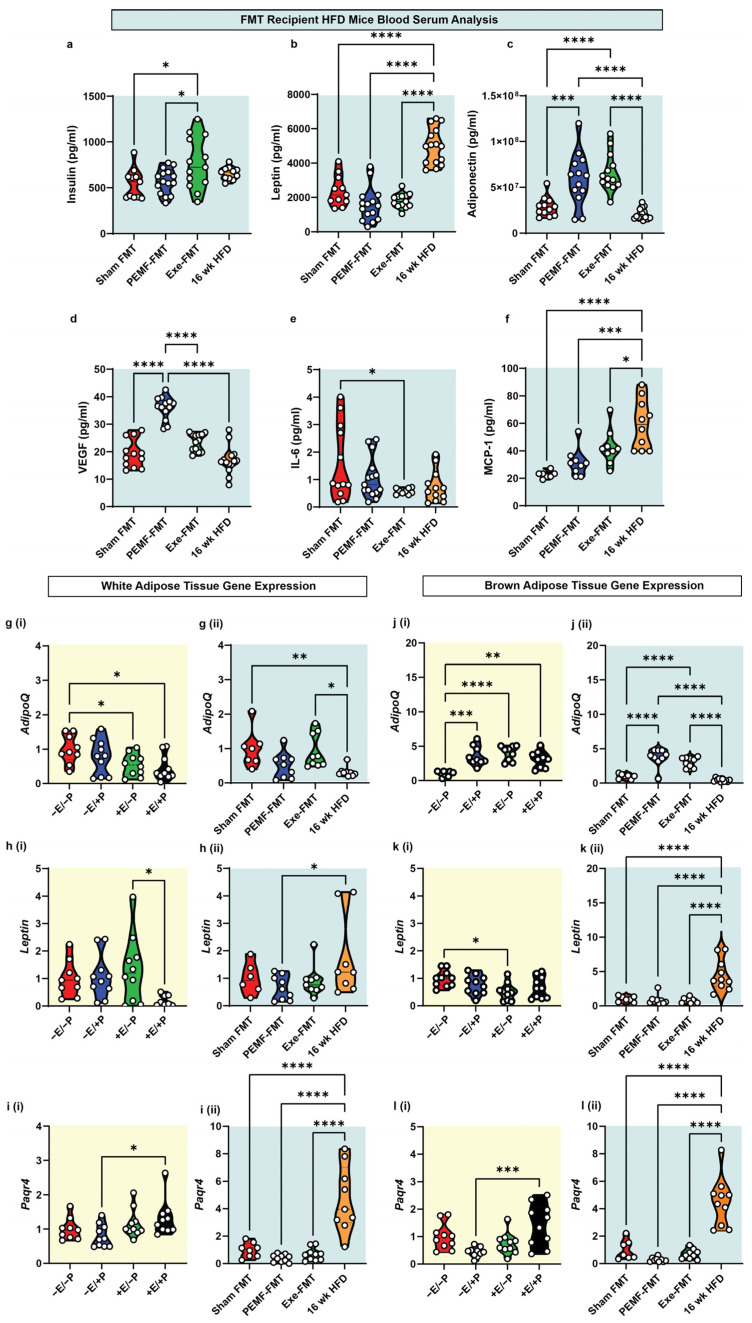
Adipokine and adipokine-related gene expression. Violin plots showing the absolute abundance of (**a**) insulin, (**b**) leptin, (**c**) adiponectin, (**d**) VEGF-A, (**e**) IL-6 (multiplex), and (**f**) MCP-1 (ELISA) from the blood plasma of recipient mice. WAT gene expression of (**g**) *AdipoQ*, (**h**) *Leptin*, and (**i**) *Paqr4* of (**i**) FMT donor and (**ii**) FMT recipient mice. BAT gene expression of (**j**) *AdipoQ*, (**k**) *Leptin*, and (**l**) *Paqr4* of (**i**) FMT donor and (**ii**) FMT recipient mice. The white dots in the violin plots represent individual animals of n = 6–10 per group. Statistical analysis was determined by a one-way ANOVA with Sidak’s multiple comparisons tests (* *p* < 0.05, ** *p* < 0.01, *** *p* < 0.001, and **** *p* < 0.0001). (E = exercise; P = PEMF).

### 2.7. Modulation of Gut Firmicutes/Bacteroidetes and Deferribacteres/Bacteroidetes Ratios

Changes in the gut microbiota were evaluated using 16S rRNA gene sequencing of the fecal and cecal matter. In both the FMT donor and FMT recipient mice, the predominant microbial phyla were *Firmicutes*, *Deferribacteres*, *and Bacteroidetes* (Figure 8(ai,aii)). An elevated *Firmicutes*-to-*Bacteroidetes* (F/B) ratio is linked with metabolic disorders and predictive of adiposity [43]. Consistent with a previous study [15], significant reductions in the F/B ratio were found in the PEMF-exposed donor mice (−E/+P; F/B = 1.4) and in the mice with combined PEMF and exercise (+E/+P; F/B = 1.3), relative to the control mice (−E/−P; F/B = 1.9) (Figure 8(bi)). On the other hand, exercise alone (+E/−P) did not significantly alter the F/B ratio relative to the control mice (−E/−P). In the FMT recipients, both Exe-FMT and PEMF-FMT resulted in a modest reduction in the F/B ratio, 2.5 and 2.6, respectively, compared to the 16-week HFD mice (3.5). Sham FMT did not change the F/B ratio (3.4) relative to the 16-week HFD mice.

An elevated ratio of *Deferribacteres* to *Bacteroidetes* (D/B) is indicative of obesity, inflammatory bowel disease, and colorectal cancer [59,60,61]. The *Deferribacteraceae* family is composed primarily of mucus-dwelling commensals. One member species, *Mucispirillum schaedleri*, is typically beneficial but can initiate damaging immune responses upon opportunistic aberrant expansion [62]. In exercised mice (+E/−P), the D/B ratio trended higher (~10 times; D/B = 0.21) than in the control mice (−E/−P; D/B = 0.02) (Figure 8(ci)), whereas direct PEMF exposure (−E/+P) did not alter the D/B ratio (0.07). When combined with exercise, however, PEMF exposure prevented the exercise-induced elevation in the D/B ratio (+E/+P; D/B = 0.07). All the FMT recipient groups exhibited similar D/B ratios (Figure 8(cii)).

### 2.8. Modulation of Hepatic Lipidome

Lipidomic analysis of mouse liver tissue using LC/GC-MS revealed significant alterations in the relative ratios of lipids following PEMF exposure or exercise intervention (Figure 9a,b). PEMF exposure led to substantial upregulations (with the higher relative ratios shaded in green) of individual phosphatidylcholine (PC), phosphatidylethanolamine (PE), and phosphatidylglycerol (PG) species. The upregulation of hepatic PC was accompanied by increased levels of sphingomyelins (SMs) and a corresponding decrease in total ceramide species (ceramides, CERs; dihydroceramides, DCERs; hexosylceramide, HCER; and lactosylceramide, LCER). The ceramide species were differentially modulated according to chain length (CERs, Figure 9a). Both PEMF and exercise downregulated long-chain ceramides (CERs; C14:0–C20, gray/black shading), while PEMF exposure alone upregulated very long-chain ceramides (CERs; C22–C26, green shading) (Figure 9a). The statistical significances between the groups for CERs C14–C20 and CERs C22–C24 are given in Appendix A. Cholesteryl esters (CEs) were elevated in both the PEMF and exercise groups, whereas glycerol lipids, including triacylglycerol (TAG), diacylglycerol (DAG), and monoacylglycerol (MAG), were downregulated in the PEMF-treated samples relative to the exercise or control samples.

The hepatic phospholipid profile of the PEMF-FMT recipients closely resembled that of the PEMF-treated donors, showing significant upregulations for PC, PE, and PG (Figure 10a,b) relative to Sham FMT, Exe-FMT, or mice on a 16-week HFD. Total ceramides and sphingomyelins trended higher in the PEMF-FMT and Exe-FMT mice compared to the Sham FMT mice (Figure 10b). Notably, PEMF-FMT showed the greatest elevation in total CERs, driven primarily by an increase in very long-chain ceramide species (C22–C26), including very long-chain DCERs and HCERs (see Figure 10a and Appendix A for statistical analyses of the (C22–C26) ceramides between groups). The increases in ceramides observed in the PEMF-FMT group coincided with increases in the sphingomyelin species, particularly very-long-chain sphingomyelin species (C22:0–C26:0) (Figure 10a and Appendix A). Cholesteryl esters (CEs) and TAG were elevated in both PEMF-FMT and Exe-FMT groups compared to the Sham FMT group but remained lower than those in the 16-week HFD group. Consistent with an enhanced synthesis of sphingomyelin from PC and ceramides, DAG and MAG levels remained elevated in the PEMF-FMT mice.

## 3. Discussion

The global incidence of obesity has tripled since 1975 to now over 2 billion adults [63]. The root causes of obesity are multifactorial and are contributed to by combinations of sedentary lifestyles, excessive caloric intake, environmental and nutritional contaminants, and genetic predisposition [63,64]. The ectopic accumulation of adipose tissue into visceral and deep peripheral tissues results in muscular, abdominal, and hepatic adiposity; elevated serum triglycerides; system-wide inflammation; insulin resistance; and dysregulated blood glucose levels. Obesity also undermines musculoskeletal function, disrupts hormonal balance, and promotes gut microbiome dysbiosis, which further augments metabolic disruption. The combination of all these factors greatly increases the risk of developing type-2 diabetes and cardiovascular diseases [65,66,67].

### 3.1. Modulation of Adipose, Bone, and Muscle Phenotypes

This current study demonstrated that PEMF treatment paralleled the effects of exercise in promoting the WAT and BAT gene expression involved in mitochondrial biogenesis and glucose metabolism, while reducing the gene expression related to cytotoxic ceramide synthesis. These changes were associated with a shift in the metabolic characteristics of the WAT towards a more thermogenic and less inflammatory brown adipose tissue (BAT) phenotype [31]. Specifically, direct PEMF exposure increased the transcriptional activity of *Pgc1a* required for mitochondriogenesis [15] and upregulated the expression of the key regulators of brown adipose differentiation, *Prdm16* and *Cebpa* [68], as well as mitochondrial biogenesis and respiratory activity, *Cox7a1* and *Ucp1*, characteristic of BAT [31]. Remarkably, the obese recipient mice receiving FMT from the PEMF-treated mice recapitulated the elevated donor levels of *Pgc1a* in the WAT and BAT. Moreover, the microbiome-mediated effects were more pronounced in the BAT than in the WAT, reproducing the elevated mitochondrial (*Pgc1a* and *Cox7a1*) and metabolic (*Glut4* and *Rpl23*) donor levels. These findings indicate that the transfer of the gut microbiome is capable of partially recreating adipogenic improvements in recipients and that PEMF exposure is a potent manner of conditioning the donor microbiome to this end.

In principle, a healthy aging strategy would encompass methods to maintain bone density and skeletal muscle mass in later life. In actual fact, however, normal aging is more commonly associated with the gradual loss of bone and muscle and increased ectopic adiposity, resulting in sarcopenia, osteopenia, and osteoporosis [69,70]. While weight-bearing exercises support bone health, they are challenging for the frail and non-ambulant to undertake [71]. PEMF treatment was previously shown to promote oxidative muscle development and improve adipose metabolism in mice [15,29] and humans [15,29]. In mice, where it was explicitly examined, this same PEMF paradigm was associated with increased mitochondrial fatty acid oxidation and similar microbiome shifts as those observed in the present study [15,29]. Novelly, here it is shown that Exe-FMT, and to a lesser degree PEMF-FMT, recipient mice exhibited increased expression levels of muscular *Pgc1a* and the associated mitochondrial and oxidative genes, *Nrf2*, *Sirt1*, and *Tfeb*, implicated in oxidative muscle development and resilience to oxidative damage [72]. The muscle PGC-1α-Nrf2 axis is crucial to establishing cellular antioxidant defenses, muscle regeneration, and mitochondrial biogenesis [73,74]. *Nrf2* expression was significantly elevated in both the PEMF-FMT and Exe-FMT recipients, aligning with previous evidence showing that the gut microbiota is intimately involved in modulating the systemic inflammatory status [7,75,76]. The shift in type IIA/IIB muscle fiber ratio induced by PEMF-FMT in the HFD recipient mice may reflect a transition towards a more oxidative muscle phenotype of anti-inflammatory character [72,77].

Muscle–adipose crosstalk also encroaches upon bone health [54]. An increase in bone mineral density was observed with the PEMF exposure of donor mice that was later evident in the HFD-fed recipient mice following FMT from these same donors (Figure 5), an indication of microbiome transmittance with osteogenic ramifications. Specifically, a history of an HFD was shown to compromise bone maintenance (Figure 5(ciii)), which could be partially recuperated with FMT from PEMF-treated donors (Figure 5(cii)) exhibiting elevated bone density (Figure 5(ci)). Interventions that enhance BMD by ≥1.0% are considered clinically significant in humans [78]. Notably, PEMF therapy demonstrated increases in BMD in both directly exposed and PEMF-FMT mice that exceeded this cutoff value, or +1.2% and +1.8%, respectively, foreshadowing the therapeutic potential of PEMF therapy for managing bone-related disorders.

The development and activation of oxidative muscle is essential in establishing metabolic health via the selective actions of its secretome [24,79]. Prior studies in cells [25,26,27], animals [15,26], and humans [28,29] have shown that this PEMF intervention enhances the oxidative capacity of skeletal muscle. Consistent with improvements in oxidative muscle development, recent human trials have highlighted the potential of this PEMF paradigm to promote muscle–adipose adaptations [15,28,29]. PEMF therapy also produced significant improvements in body composition in elderly individuals that included reductions in total body and visceral fat, increases in skeletal muscle mass, and enhanced mobility function after 2–3 months of intervention [29]. Additionally, PEMF therapy was shown to reduce serum ceramide levels in post-surgical patients, offsetting the lipotoxic ramifications of physical inactivity [28].

Given that obesity-related disruptions of the gut microbiome have been implicated in human inflammatory and metabolic disorders [65,66,67], FMT has been discussed as a potential method for the management of human metabolic complications, including insulin resistance, lipid imbalances, and central obesity [46,80]. Gut–muscle crosstalk at the mitochondrial level may explain the enhancements of muscle oxidative capacity, adipose browning, hepatic lipid profiles, and bone density observed following FMT from PEMF-exposed mice. The defining features of the oxidative muscle phenotype are augmented mitochondrial respiration and elevated PGC-1α expression and transcriptional activity [30]. As our magnetic field treatment activates mitochondrial respiration and upregulates PGC-1α expression [25], muscle oxidative capacity is also enhanced following periods of PEMF treatment [15]. Evidence also exists that microbiomal metabolites can serve as substrates for muscle mitochondrial respiration [9,49,81] and that microbiomal species can be transferred from a donor to recipients with the transmittance of the donor’s characteristics [45]. It is thus possible that the metabolic features of the donor may be transferred to the FMT recipients via an interaction with the microbiome species transferred from the donor to the recipient. A summary of this study’s results is provided in Table 1 (located at the end of the discussion).

### 3.2. Modulation of Blood Adipokines, Angiogenic Factors, and Glucose Handling

Obesity disrupts insulin signaling and glucose metabolism by shifting the emphasis of the adipose secretome to a pro-inflammatory status [54,82]. Adipokines, such as adiponectin and leptin, help orchestrate the cytokine crosstalk between the muscle, adipose, and bone to establish systemic energy homeostasis and metabolic balance [54]. HFDs have been shown to decrease adiponectin and increase leptin in the WAT and blood [82,83] in correlation with weight gain and disruptions in glucose handling [84]. In an opposing manner, here it is shown that the cessation of the HFD normalized the serum leptin levels relative to mice maintained on the HFD (Figure 6b) and, moreover, that PEMF-FMT and Exe-FMT significantly increased serum adiponectin levels (Figure 6c). Adiponectin enhances fatty acid oxidation and insulin sensitivity to forestall the onset of type 2 diabetes and cardiovascular diseases [54]. Expectedly, exercise increases plasma adiponectin levels with a corresponding drop in leptin levels in prediabetic and diabetic patients [85]. While both PEMF-FMT and Exe-FMT elevated the adiponectin levels in the recipient HFD mice, PEMF-FMT outperformed Exe-FMT in terms of blood glucose control. Finally, the WAT and BAT *Paqr4* levels were negatively correlated with *AdipoQ* and positively correlated with *Leptin* (Figure 6). PAQR4 regulates adipose ceramide levels; the dysregulation of PAQR4 leads to an accumulation of lipotoxic ceramide species, particularly long-chain ceramides [58,86], correlated with the increased risks of developing metabolic and hepatic diseases [58,86]. Accordingly, reductions in long-chain ceramides were observed in hepatic lipid profiles (see Section 3.4*. Hepatic lipidome*).

VEGF-A stimulates angiogenesis during tissue regeneration and growth. VEGF-A is typically released from exercising muscles to stimulate muscle microvascularization, growth, and repair [87,88]. The increase in plasma VEGF-A observed in the PEMF-FMT recipient mice, but not in the Exe-FMT, suggests a preferential enhancement in muscular microvascularization induced by PEMF therapy. Notably, diabetic patients exhibit lower plasma VEGF-A [89], whereas the overexpression of VEGF-A “beiges” the WAT and enhances the expression of PGC-1α and UCP-1 in the BAT [89,90]. Consistent with these findings, the PEMF-exposed donor and PEMF-FMT recipient mice displayed increased *Pgc1a* expression in the WAT, rendering it more BAT-like (Figure 2a). Both PEMF treatments also increased *Pgc1a* expression in the BAT (Figure 3a), recapitulating a known thermogenic response [91]. The microbiome-mediated VEGF-A upregulation observed in the PEMF-FMT recipient mice (Figure 6d) thus aligns with the observed improvements in metabolic health and adipose tissue remodeling (Figure 2).

### 3.3. Firmicutes/Bacteroidetes (F/B) and Deferribacteres/Bacteroidetes (D/B) Ratios

This study also investigated the effects of PEMF exposure on gut microbiota using cecal and fecal samples as proxies [92]. PEMF exposure reduced the *Firmicutes/Bacteroidetes* (F/B) ratio, which was previously correlated with a healthier phenotype [15]. This shift occurred independently of alterations in α-diversity or short-chain fatty acid production (Appendix A) [50,93,94]. An elevated F/B ratio is predictive of an elevated body mass index (BMI) and obesity [93,95]. Notably, PEMF-FMT and Exe-FMT effectively lowered the F/B ratio in obese recipient mice (Figure 8), despite exercise alone having no impact on the F/B ratio. The *Deferribacteres* are pro-inflammatory bacteria that thrive in low-oxygen environments and contribute to metabolic disease. Classified under the phylum *Deferribacteres*, the *Mucispirillum* species is linked to pregnancy-related stress, social stress, and HFD-induced metabolic disorders in mice [96,97]. In humans, *Mucispirillum* species are linked to inflammatory bowel disease and Parkinson’s disease, particularly in those individuals with sub-inflammation and elevated TNF-α and IFN-γ levels [62]. Exercise increased the *Deferribacteres/Bacteroidetes* (D/B) ratio, indicative of exercise-induced inflammation, whereas PEMF exposure mitigated this effect. In accordance, excessive exercise stress may increase gut epithelial permeability and dysbiosis [66]. These results underscore the unique therapeutic potential of this PEMF paradigm to modulate gut microbiota for the mitigation of inflammatory and metabolic disturbances with minimal stress.

### 3.4. Hepatic Lipidome

Hepatocyte mitochondrial membrane integrity is crucial for liver function [98]. Exercise effectively mitigates the adverse consequences of an HFD over the liver mitochondrial phospholipidomic profile in a rat model of non-alcoholic steatohepatitis (NASH) [98]. Specifically, exercise sustains the hepatic levels of phosphatidylinositol (PI), cardiolipin (CL), and the ratio of phosphatidylcholine/phosphatidylethanolamine (PC/PE) essential for mitochondrial function [99]. Phosphatidylglycerol (PG) is the glycerophospholipid precursor for CL, a mitochondrial membrane phospholipid required for mitochondrial biogenesis [99]. PG deficiencies result in CL deficiencies, mitochondrial dysfunction, and reductions in ATP production. Accordingly, this present study showed that exercise and PEMFs increased PC, PE, and PG (Figure 9), foreshadowing the previous observations of enhanced mitochondrial biogenesis and efficiency by PEMF treatment [15]. Direct PEMF exposure also increased the hepatic levels of PI and PS (phosphatidylserine), while reducing the levels of the lysophospholipids (LPLs), LPC (lysophosphacholine) and LPE (lysophosphaethanolamine) (also see Appendix A). The reductions in the LPLs suggest that exercise and PEMFs enhance phospholipid remodeling by promoting the formation of more complex phospholipid species. LPC was downregulated by both PEMFs and exercise and is known to exhibit pro-inflammatory effects [100] and contribute to the onset of metabolic disorders such as diabetes [101].

Unlike exercise, both PEMF exposure and PEMF-FMT upregulated hepatic sphingomyelin, which, evidence suggests, is protective against steatosis, steatohepatitis, fibrosis, insulin resistance, and cancer [102,103]. This is important because sphingomyelin prevents the accumulation of the ceramides and glucosylceramides. On the other hand, PEMF exposure mimicked exercise in reducing overall hepatic ceramides, which should also alleviate hepatic steatosis and insulin resistance [104]. Hepatic ceramides contribute to hyperglycemia and insulin resistance by inhibiting the action of Akt/PKB, disrupting insulin’s ability to suppress hepatic glucose production and inhibiting glycogen synthesis [105]. An accumulation of hepatic ceramides also promotes lipid uptake and storage into hepatocytes, exacerbating hepatic steatosis and further impairing glucose metabolism.

Ceramide species cytotoxicity is largely determined by their structure. Deleting dihydroceramide desaturase 1 (DES1) in the liver resolved hepatic steatosis and improved insulin sensitivity in diet-induced obese mice by increasing the dihydroceramide (DCER) levels that decreased ceramide species with a critical double bond [106]. Long-chain C14-C20-ceramides, especially C16:0-ceramide, are strongly associated with insulin resistance. Elevated levels of long-chain C16:0-ceramide in the liver have been shown to inhibit beta-oxidation, exacerbating liver steatosis and insulin resistance in mice [86,107]. Consistent with these findings, both direct PEMF exposure and exercise reduced hepatic long-chain ceramides (C14–C20), in alignment with improved insulin sensitivity and reduced systemic inflammation (Figure 7 and Figure 9). On the other hand, very long-chain ceramides, such as C24:0-ceramide, are protective against HFD-induced obesity and glucose intolerance [107]. Very long-chain ceramides (C22–C26) are associated with enhanced cellular membrane integrity and improved lipid metabolism, promoting fatty acid breakdown and utilization. Accordingly, the serum ratio of long-chain to very long-chain (C16:0/C24:0) ceramides has been proposed as a valuable predictor for coronary heart disease, heart failure, and all-cause mortality in humans [108]. Notably, very long-chain ceramides, particularly C24:0-ceramide, were increased in the PEMF-treated mice. A shift towards very long-chain ceramides was also achieved by PEMF-FMT of HFD-fed recipient mice; very long-chain C22–C26 ceramides were upregulated in association with an improvement in glucose handling (Figure 7). By contrast, the mice maintained on an HFD showed significant increases in long-chain ceramides (C14–C20) and reductions in very long-chain ceramides (C22–C26), associated with metabolic dysfunction. In summary, both PEMF exposure and PEMF-FMT of HFD mice showed increases in very long-chain C22-C26 ceramides, supporting the potential for PEMF therapy to improve and restore metabolic balance by promoting a healthier C16:0/C24:0 ratio. These results agree with previous human data demonstrating that weekly PEMF therapy modulated ceramide levels in humans [28].

Neutral lipids, such as cholesteryl esters (CEs) and triacylglycerol (TAG), play vital roles in membrane formation, lipoprotein trafficking, lipid detoxification, and energy storage. The conversion of cholesterol to cholesteryl esters helps maintain free cholesterol balance and reduces the risk of cholesterol-induced toxicity [109]. PEMF exposure and exercise increased hepatic cholesteryl esters, which reportedly enhances the capacity of HDL to remove excess cholesterol from peripheral tissues [110]. Cholesteryl ester levels were also normalized following PEMF-FMT and Exe-FMT of HFD-fed recipient mice, implicating the microbiota in the response.

PEMF exposure significantly downregulated triacylglycerol (TAG), as well as its metabolic intermediates diacylglycerol (DAG) and monoacylglycerol (MAG), recapitulating a metabolic profile that has been correlated with improved outcomes in NAFLD [111]. By contrast, exercised mice showed an elevation in DAG and MAG (Figure 9), possibly reflecting exercise-induced lipolysis of these intermediates into free fatty acids, which can then be utilized by the mitochondria as energy substrates [112]. These distinct, yet complementary, hepatic lipidomic signatures may indicate that PEMF exposure enhances lipid clearance, whereas exercise stimulates lipid mobilization; ultimately, both pathways serve to improve metabolic health [113]. Moreover, FMT from either PEMF-treated or exercised donors replicated the PEMF-induced reduction in TAG in the HFD-fed recipient mice (Figure 10). Notably, Exe-FMT showed the greatest reductions in DAG/MAG, consistent with the described microbiomal transmissible effects of exercise and diet on obesity [114]. While Sham FMT (the saline control) also showed reductions in hepatic lipid content following HFD (Figure 8), FMT from both PEMF and exercised mice demonstrated greater improvements in adipose/muscle gene expression (Figure 2 and Figure 4), bone density (Figure 5), adiponectin (Figure 6), and the F/B ratio (Figure 8). These results reveal the multi-tissue therapeutic potential of muscle-mediated microbiome modulation produced by exercise or PEMF interventions [49].

**Table 1 ijms-26-05450-t001:** Summary of biological effects of direct PEMF exposure and PEMF-FMT in mice.

Figure	Objective	Analysis/Statistical Test	Findings		Ref.
			Donor Mice	FMT Recipient HFD Mice	
1	GROWTH RATE	Weight	No weight difference between groups (n = 10 per group)	Continual weight increment in HFD mice. PEMF-FMT showed the greatest stabilization of weight (n = 12–15 per group)	[84]
2	WHITE ADIPOSE TISSUE Expression of Metabolic Genes	qPCR One-way ANOVA (Sidak’s test)	PEMF vs. control * Significant upregulation: *Pgc1a, Glut4, Rpl23*, and *Prdm16* Modest upregulation: *Cebpa* and *Ucp1* (n = 6–10 per group) * *p* < 0.05	PEMF-FMT vs. Sham FMT * Significant changes: *Pgc1a* upregulation *Cox7a1* downregulation (n = 7–10 per group) * *p* < 0.05	[15,31,68,73,74]
3	BROWN ADIPOSE TISSUE Expression of Metabolic Genes	qPCR One-way ANOVA (Sidak’s test)	PEMF vs. control * Significant upregulation: *Pgc1a, Cox7a1, Glut4, Rpl23*, (n = 6–10 per group) * *p* < 0.05	PEMF-FMT vs. Sham FMT * Significant upregulation: *Pgc1a, Cox7a1, Glut4, Rpl23*, (n = 7–10 per group) * *p* < 0.05	
4	MUSCLE Expression of Metabolic Genes	SOLEUS qPCR One-way ANOVA (Sidak’s test)	Not applicable Note: PEMF was previously shown to upregulate *Pgc1a* expression [15].	PEMF-FMT vs. Sham FMT * Significant effect: *Nrf2* upregulation Modest effect: *Ppara*, *Pgc1a*, *Tfeb*, and *Sirt1* Note: Exercise FMT significantly upregulated gene targets compared to Sham FMT (n = 3–5 per group) * *p* < 0.05	[15,30,53,73,77]
	MUSCLE Protein Expression	EDL Western Blot One-way ANOVA (Sidak’s test)	Not applicable	PEMF-FMT vs. Sham FMT Modest elevation: PGC-1α and Type IIA fibers (n = 4–6 per group)	
4 & 5	BONE DENSITY	Micro-CT One-way ANOVA (Sidak’s test)	PEMF vs. control * Significant bone fortification: Cortical: thickness, volume, and BMD Trabecular: thickness and number (n = 10 per group) * *p* < 0.05	PEMF-FMT vs. 16 wk HFD mice * Significant bone fortification: Cortical: volume and BMD Trabecular: thickness, number, and percentage Note: Cortical BMD was also significantly higher in Exercise-FMT vs. HFD, but not in Sham-FMT vs. HFD. (n = 8–10 per group) * *p* < 0.05	[54,78]
6	BLOOD BIOMARKERS Metabolic Biomarkers	ELISA and Multiplex assay One-way ANOVA (Sidak’s test)	Not applicable	PEMF-FMT vs. Sham FMT * Significant upregulation: Adiponectin and VEGF Modest downregulation: Leptin Note: Exercise-FMT significantly increased insulin and adiponectin levels compared to Sham FMT. (n = 10–15 per group) * *p* < 0.05	[82,83,87,88,91]
	WHITE ADIPOSE TISSUE Expression of Energy Homeostatic Genes	qPCR One-way ANOVA (Sidak’s test)	PEMF vs. control No observable change: *AdipoQ*, *Leptin*, and *Paqr4* (n = 9–10 per group)	PEMF-FMT vs. Sham FMT Modest downregulation: *AdipoQ*, *Leptin*, and *Paqr4* (n = 6–10 per group)	
	BROWN ADIPOSE TISSUE Expression of Energy Homeostatic Genes	qPCR One-way ANOVA (Sidak’s test)	PEMF vs. control * Significant upregulation: *AdipoQ* (n = 9–10 per group)	PEMF-FMT vs. Sham FMT * Significant upregulation: *AdipoQ* (n = 6–10 per group)	
7	INSULIN SENSITIVITY Glucose Tolerance Test	IPGTT One-way ANOVA (Sidak’s test)	Not applicable	PEMF-FMT vs. Sham FMT Prompt normalization of blood glucose: * Significant difference at 60 min and 90 min post glucose challenge (n = 12–15 per group) * *p* < 0.05	[82,83,84]
8	MICROBIOME DIVERSITY AND COMPOSITION *Firmicutes*-to-*Bacteroidetes* (F/B) ratio.	16S rRNA gene sequencing One-way ANOVA (Sidak’s test)	PEMF vs. control * Significant effect: Lower F/B Ratio (1.4 vs. 1.9) (n = 4–8 per group) * *p* < 0.05	PEMF-FMT vs. Sham FMT Modest effect: Lower F/B Ratio (2.6 vs. 3.4) (n = 4–8 per group) * *p* < 0.05	[93,95]
9 & 10	HEPATIC LIPIDS	Lipidomics One-way ANOVA (Tukey’s test)	PEMF vs. control Significant changes: Elevation in PC, PE, PG, sphingomyelins, and cholesteryl esters Reduction in TAG, DAG, and MAG Ceramides: Reduction in long-chain (C16–C20) and elevation in very long-chain ceramides (C22–C26). This distinct expression pattern is not observed with exercise. (n = 5 per group) * *p* < 0.05	PEMF-FMT vs. Sham FMT Significant changes: Elevation in PC, PE, PG, ceramides, sphingomyelins, cholesteryl esters, TAG, DAG, and MAG PEMF-FMT vs. 16 wk HFD Ceramides: Reduction in long-chain (C16–C20) and elevation in very long-chain ceramides (C22–C26). This distinct expression pattern is not observed with Sham FMT or Exercise-FMT. (n = 4–6 per group) * *p* < 0.05	[28,86,102,103,107,108,110,112]

## 4. Materials and Methods

### 4.1. Animal Husbandry, Food Intake, and Ethics Approval

The mice were housed in socially suitable groups under a strict regimen of 12 h of light and 12 h of darkness. All the animals received standard chow as food and water ad libitum. Weekly food consumption was measured for each intervention group. For each group, the total food intake was determined by measuring the reduction in food mass across all the cages for the mice of the same group. The average food intake per mouse was calculated by dividing the total weekly food consumption by the number of mice in the group. The Institutional Animal Care and Use Committee (IACUC) of the National University of Singapore (approval number: R19-0149) approved the overall experiment. All the rules and regulations regarding the treatment and care of laboratory animals were strictly followed while performing all the interventions.

### 4.2. Experimental Groups for Paradigm 1: Direct Magnetic Exposure

The 7-week-old male C57BL/6NTac mice (n = 48) were randomized into five different treatment groups, i.e., (1) baseline: 2 weeks of acclimatization (n = 8); (2) control (n = 10); (3) PEMF-only (n = 10); (4) exercise-only (n = 10); and (5) PEMF and exercise group (n = 10). The method of randomization was as follows: the mice were retrieved from the delivery cages used by the animal facility at random and placed sequentially in the respective cages for each experimental condition, i.e., starting from the baseline group to the PEMF and exercise group, until all the mice were allocated. In order to minimize confounding factors, the order of treatments was mixed each time the researchers performed interventions on the animals. The mice were acclimatized to their surroundings for 2 weeks before the start of the 8-week intervention. No blinding was performed at any stage of this study.

### 4.3. PEMF and Exercise Interventions

This present study employed a previously published mouse exposure paradigm [15,25]. The mice were exposed once per week to 1.5 mT PEMF for 10 min for 8 weeks alone or in combination with treadmill running (Figure 1). The mice destined for the exercise cohorts were accustomed to the treadmill by allowing them to roam on the stationary treadmill for 5 min, followed by a 10 min walk at a speed of 6 m per min two weeks prior to the commencement of the actual study intervention. The mice were then subjected to 8 weeks of rigorous treadmill training, twice weekly. The Graded Maximal Exercise Test (GXTm) protocol was employed with modest modifications [15,25]. Treadmill running was stimulated in mice with auditory stress (noise) using the noise of crumpling paper towels below the running belt, instead of the usual electrical shock (Columbus Instruments, Colombus, OH, USA). The overall exercise regime was as follows: (speed, duration, grade)—(0 m/min, 3 min, 0°), (6 m/min, 2 min, 0°), (9 m/min, 2 min, 5°), (12 m/min, 2 min, 10°), (15 m/min, 2 min, 15°), (18, 21, 23, 24 m/min, 1 min, 15°), and (+1 m/min, each 1 min thereafter, 15°) until exhaustion. The mice were considered to have reached their point of exhaustion when they remained on the stationary platform for 30 sec without making any additional attempts to climb back onto the treadmill, despite the ongoing presence of auditory stimuli.

### 4.4. Collection of Fresh Fecal Pellets for SCFA Analysis

Two weeks before study termination, individual mice were placed into sterile plastic boxes and allowed to defecate ad libitum for 30 min intervals to collect fresh fecal pellets. Each session consisted of six mice housed separately in six boxes. The sample collection was conducted daily over a period of 2–3 h. For each 30 min collection interval, an average of 20 mg of fresh fecal pellets was obtained per mouse. The total daily fecal output was not measured. The collected fresh fecal pellets were snap-frozen and stored at −80 °C until further examination. The samples were sent to Duke-NUS Metabolomics Facility for short-chain fatty acid (SCFA) analysis. Briefly, the process of preparing the mouse fecal samples for analysis began with the initial steps of weighing the fecal pellet and adjusting their concentration to 100 mg/mL by adding ethanol. Afterward, the fecal samples were homogenized at 4 °C, and 100 μL aliquots of the resulting supernatant were utilized for the analysis. To prepare these samples, a 10 μL mixture of internal standards (acetate-d3, propionic acid-13C, butyric acid-13C, and valeric acid-d9) were added, followed by the addition of 100 μL of 1.0 M formic acid prepared in ultrapure water. After vortexing for 1 min, the samples were centrifuged, and a clear supernatant was collected for analysis. A Thermo Scientific Trace 1310 gas chromatography instrument that was linked to a TSQ 8000 Evo mass spectrometer was used to separate the SCFAs (Thermo Fisher Scientific, Waltham, MA, USA) using a DB-Fatwax UI column, 30 m × 0.25 mm × 0.25 µm (Agilent Technologies, Santa Clara, CA, USA). The GC method was in splitless mode, and the injector temperature was set at 250 °C. The temperature gradient ran from 40 °C to 250 °C. The MS method consisted of ionization at 70 eV, followed by the observation of numerous reactions and transitions for a total of 13 min. Peaks were identified by comparison with known standards and quantified using TraceFinder 4.1 software.

### 4.5. Dried Fecal Pellets for Fecal Microbiota Transplantation, Cecal Matter, and Adipose Tissue Collection

Dried fecal pellets were collected between weeks 7 and 8 from the mice that had received weekly PEMF therapy or exercised as mentioned before (Section 4.3
*PEMF and Exercise Interventions*). To account for three daily gavages and a 10% handling loss, approximately 800 mg of fecal pellets were collected per day over 10 days, yielding the required 20 mg of feces per gavage for an 8-week period. The collected fecal samples were immediately homogenized in anaerobic PBS to preserve microbial integrity and stored at −80 °C until use.

At the termination of this study, following CO_2_ euthanasia, the ceca and moist fecal matter were snap-frozen and stored at −80 °C. The adipose tissue from the interscapular brown and subcutaneous white inguinal regions were also stored away.

### 4.6. Experimental Groups for Paradigm 2: Fecal Matter Transplant (FMT)

The 7-week-old male C57BL/6NTac mice (n = 81) were randomized into 5 groups, i.e., (1) 4 weeks of acclimatization (n = 10); (2) 8 weeks of a high-fat diet (HFD) (n = 11); (3) Sham FMT (n = 15); (4) 16 weeks of an HFD (n = 15); (5) PEMF-FMT (n = 15); and (6) Exercise-FMT (n = 15). The method of randomization and the no-blinding protocol were identical to those of the experimental groups for paradigm 2 in Section 4.2. The mice were acclimatized to their surroundings for 4 weeks before the start of the 16-week study.

### 4.7. Antibiotic Administration, High-Fat Diet (HFD), and FMT Intervention

After acclimatization, all the mice were given an HFD (Teklad 06415, Envigo, Cumberland, VA, USA) to induce obesity and hyperglycemia for 8 weeks. During the 8-week HFD, the mice were given an antibiotic cocktail through oral gavage biweekly from week 5 to week 8. The antibiotic cocktails (Merck, Darmstadt, Germany) included 100 μg/mL neomycin, 50 μg/mL vancomycin, 50 μg/mL imipenem, 100 μg/mL metronidazole, 50 μg/mL streptomycin, and 100 U/mL penicillin [115]. One group, i.e., the 16-week HFD group, continued with the HFD for another 8 weeks. The FMT intervention through oral gavage was administered to the animals after 8 weeks of the HFD. The FMT groups were (1) Sham FMT; (2) PEMF-FMT; and (3) Exercise-FMT. The FMT-receiving HFD mice were placed back on a standard chow diet (SC) for the remaining 8 weeks of this study. Oral gavage of the FMT was performed on a thrice-weekly basis for a total of 8 weeks. The fecal pellets were collected and processed from paradigm 1 (Direct Magnetic Exposure) (see Section 4.5
*Dried fecal pellets for fecal microbiota transplantation*, *cecal matter, and adipose tissue collection*). For each gavage, 20 mg of fecal matter was resuspended in 400 µL of anaerobic PBS prior to administration. Sham FMT was given as a gavage of 400 µL phosphate-buffered saline (PBS).

### 4.8. Intraperitoneal Glucose Tolerance Test (IGTT)

At the end of the FMT treatment, the mice were assessed across all the groups for glucose tolerance using the intraperitoneal test (IPGTT). After 12 h of food abstinence, the mice received an intraperitoneal injection of 0.9% saline containing 2 g of glucose (Sigma-Aldrich, St. Louis, MI, USA) per kg of body weight. A glucometer (Roche Diagnostics, Rotkreuz, Switzerland) was used to analyze the blood glucose levels before and at 0, 10, 20, 30, 60, 90, and 120 min after the glucose injection [116].

### 4.9. Blood Plasma and Multiplex Analysis

Blood was collected via cardiac puncture after the sedation of the mice using isofluorane. The blood plasma was separated using EDTA-coated tubes and was centrifuged at 2000× *g* for 10 min at 4 °C. The plasma (supernatant) was collected and stored at −80 °C until further analysis. The blood plasma was analyzed using a customized mouse-specific multiplex kit (Bio-Rad Laboratories, Hercules, CA, USA) to simultaneously quantify the presence of the following analytes: adiponectin, insulin, leptin, interleukin 6 (IL-6) and vascular endothelial growth factor (VEGF). The immunoassay procedure was performed according to the manufacturer’s workflow. Briefly, all the samples (25 μL/sample/well) were incubated with fluorescent-coded magnetic beads pre-coated with the respective antibodies, followed by the incubation of the sample–antibody–bead complexes with biotinylated detection antibodies and Streptavidin-PE. The samples were washed after each incubation step before the sample–antibody–bead complexes were resuspended in sheath fluid for acquisition on the Luminex platform (Luminex LX200, Luminex Corporation, Austin, TX, USA) using the xPONENT^®^ 4.0 (Luminex, Luminex Corporation, Austin, TX, USA) software. Standard curves were generated with a 5-PL (5-parameter logistic) algorithm, reporting values for mean fluorescence intensity (MFI). The NetMFI values were normalized to the mean value of the blank control (background) before the analyte concentration values were extrapolated from the respective analyte standard curves.

### 4.10. Microbiome Profiling of Stool Samples Using 16S rRNA Gene Sequencing

Total bacterial DNA was extracted from the cecal and fecal samples using the DNeasy PowerSoil Kit (Qiagen, Hilden, Germany) after the samples were thawed and weighed per the manufacturer’s instructions. The overall yield and purity of the DNA were determined using Nanodrop One (Thermo Fisher Scientific, Waltham, MA, USA). The extracted DNA samples were outsourced to NOVOGENE PTE. Ltd. (Singapore) for the 16S rRNA gene sequencing. The analysis was performed on the Illumina HiSeq 2000 platform to generate 250 base pair paired-end raw reads. To examine the microbial community makeup of the cecal and fecal samples of the mice belonging to the various treatment groups, 4 mice each were selected from every treatment group in the direct PEMF investigation, and 6 mice were selected from the FMT study. This was achieved by 16S rRNA gene amplicon sequencing targeting the V4 region of bacterial 16S rRNA genes using the 515F (5ʹ-GTGCCAGCMGCCGCGG-3ʹ) and 907R (5ʹ-CCGTCAATTCMTTTRAGTTT-3ʹ) primers.

### 4.11. MicroCT Analysis of the Tibia

After euthanasia, the right hind leg of each mouse was preserved in 70% ethanol. The tibia was disarticulated, and the connective tissues were painstakingly removed using tweezers. After this, the bone was wrapped in a small piece of tissue paper dipped in 70% ethanol to prevent moisture loss. The individual samples were then positioned inside a cylinder-shaped Styrofoam holder, which was then secured to the sample bed of the Skyscan 1176 small animal micro-CT scanner (Bruker BioSpin, Kontich, Belgium). The following parameters were used on the Skyscan 1176 control program to scan the right tibia: source voltage = 50 kV, source current = 500 µA, filter = 0.5 mm Al, image pixel size = 8.81 µm, exposure = 870 ms, rotation step = 0.3°, and frame averaging = 1. The following parameters were utilized for the reconstruction that was carried out with the NRecon program v2.2.2 (provided by Bruker BioSpin): smoothing = 1, ring artifact correction = 5, beam hardening correction = 32%, and the range of CS to image conversion = 0–8. During the image analysis process, the Data-viewer software v1.7 from Bruker BioSpin was utilized to align all the bone samples, such that their long axes matched those of the bones and image volumes. This was performed to assist in the selection of the ROI. An analysis of the images was carried out using Ctan software v1.23.0.2. A 100-slice core section of the tibia was selected for the cortical bone examination. The ROI tool in Ctan was used to select the cortical bone region manually. A binary cortical bone image was produced using a lower gray threshold of 120 and an upper gray threshold of 255, and then 3D analysis was carried out. A specific growth plate slice was determined by inspecting the image slices beginning at the proximal end of the tibia to study the trabeculae bone. The next 100 slices were selected for the trabeculae bone study, skipping the first 50. To achieve this, we used Ctan’s ROI tool to manually choose the trabeculae bone region. A 3D analysis was applied to a binary trabeculae bone picture with a lower gray threshold of 70 and an upper gray threshold of 255.

### 4.12. Liver Homogenization and Lipidomics Profiling

The homogenization solvent was prepared by chilling a 50% aqueous acetonitrile solution containing 0.3% formic acid. The liver tissue sample was removed from dry ice. To 50 mg of liver tissue, 950 µL of the chilled homogenization solvent was added to achieve a concentration of 50 mg wet weight per mL of homogenate. The tissue was homogenized on ice using the Precellys Evolution tissue homogenizer (Bertin Technologies, Montigny-le, France). After homogenization, the sample was either refrozen on dry ice or aliquoted for further analysis before refreezing. For lipid profiling, 25 µL of liver homogenate, 283 µL of deionized water, 581 µL of methanol (Thermo Fisher Scientific, Waltham, MA, USA), and 262 µL of dichloromethane (Sigma-Aldrich, St. Louis, MI, USA) were added to a 2.0 mL microcentrifuge tube. Once the mixture was a single phase, 25 µL of the Lipidyzer internal standard mix (SCIEX, Marlborough, MA, USA) was added, vortexed and left to stand for 30 min at room temperature. A further 291 µL of deionized water and 262 µL of dichloromethane were added and the tube was centrifuged at 1710× *g* for 10 min at 4 °C. The lower organic layer was transferred to a new microcentrifuge tube, evaporating the solvent with a nitrogen blower. The dried extract was reconstituted with 100 µL of methanol. The reconstituted lipid solution was analyzed using a liquid chromatography-mass spectrometry (LC-MS) system, employing an Agilent 1290 Infinity II LC system and a Waters XBridge Amide column (4.6 mm × 150 mm, 3.5 μm) (Waters Corp, Singapore). The mobile phases used were as follows: Phase A: 1 mM ammonium acetate in 95% acetonitrile and Phase B: 1 mM ammonium acetate in 50% acetonitrile. The column was equilibrated with 100% mobile phase A. Mass spectrometry (MS) and data acquisition were performed using an SCIEX Triple Quad 5500 MS System (SCIEX, Marlborough, MA, USA). For the acylcarnitine analysis, 50 µL of each sample was spiked with 10 µL of a deuterium-labeled acylcarnitine mixture and diluted with 400 µL of methanol. After centrifugation at 13,000 rpm for 5 min at 4 °C, the supernatant was collected and derivatized with 3 M hydrochloric acid in methanol (Sigma-Aldrich, St. Louis, MI, USA), then reconstituted in 80% methanol for the LC-MS analysis. All the compounds were ionized in positive mode using electrospray ionization. Mass spectrometry (MS) and data acquisition were performed using an Agilent 6430 triple-quadrupole mass spectrometer (Agilent Technologies, Santa Clara, CA, USA).

### 4.13. RNA Extraction and qPCR Analysis of Mouse Adipose and Muscle Samples

The total RNA was harvested from homogenized white adipose tissue (WAT) or muscle tissues using Trizol reagent (Thermo Fisher Scientific, Waltham, MA, USA). cDNA was synthesized from 0.5 μg of total RNA using the iScript cDNA Synthesis Kit (Bio-Rad Laboratories, Hercules, CA, USA). In total, 2 ng of RNA was converted to the cDNA template for the quantitative PCR. The quantification of the mRNA expression was performed using SSoAdvanced Universal SYBR Green (Bio-Rad Laboratories, Hercules, CA, USA) on the CFX Touch Real-Time PCR Detection System (Bio-Rad Laboratories, Hercules, CA, USA). The relative mRNA expression was determined using the 2^−ΔΔCt^ method, normalized to β2-microglobulin (B2m) mRNA levels. The primer pairs used are listed in Table 2 and Table 3 for the adipose and muscle tissues, respectively.

### 4.14. Analysis of Muscle Protein Abundance

Snap-frozen muscle samples were ground into a fine powder under liquid nitrogen using a metal mortar and pestle. The powdered samples were then lysed in a standard RIPA buffer [27]. The protein concentrations were determined, and 25 µg of protein from each sample were resolved by 8% SDS-PAGE. Following electrophoresis, the proteins were transferred onto a PVDF membrane. The primary antibodies used for detection were as follows: Mouse Myosin heavy chain Type I (BA-D5-c, 1:400, DSHB), Mouse Myosin heavy chain Type IIA (SC-71-c, 1:1000, DSHB), Mouse Myosin heavy chain Type IIB (BF-F3-c, 1:500, DSHB), and Mouse PGC-1α monoclonal antibody (66369-1-Ig, 1:5000, Proteintech, Singapore). Following, the secondary antibody Goat Anti-Mouse HRP (31430, 1:10,000, Thermo Fisher Scientific, Waltham, MA, USA) was used. The imaging and data analysis were performed using LICOR Odyssey FC imaging system (LI-COR Biotechnology, Lincoln, NE, USA).

### 4.15. Bioinformatics Analysis

The outputs of the paired-end reads were demultiplexed (Illumina, San Diego, CA, USA) and merged using FLASH v1.2.7. The raw tags were filtered using a Q-score of 20 and above using QIIME v.1.7.0. Using the UCHIME method, non-chimeric sequences were selected from clean tags by deleting the chimera sequences from the Gold reference database.

Representative sequences for each operational taxonomic unit (OTU) were obtained after assigning 97% similarity of 16S gene sequences by the Uparse method (v. 7.0.1001). For the taxonomic annotation, the OTUs were screened at a 0.8–1 threshold by Mothur software v1.47.1 with the SSUrRNA database of SILVA database. The abundance of OTUs was normalized using a reference sequence number of the least sequences.

Alpha diversity of the sample was described in the Chao 1 estimator, Simpson, and Shannon. Using the relative abundance of bacterial genus-level OTU data, a principal coordinate analysis (PCoA) was performed with CANOCO 5 (Microcomputer Power, Ithaca, NY, USA). Linear discriminant analysis (LDA) effect size (LEfSE) scoring was calculated, and an alpha value of the pairwise Wilcoxon test was set at 0.05 and a logarithmic LDA threshold at 2, using the Galaxy platform (a web-based bioinformatics tool).

### 4.16. Statistical Analysis

The statistical analyses and data visualization were performed using GraphPad Prism 10v10.5.0 (GraphPad Software, LA Jolla, CA, USA). Normality was assessed using the Anderson–Darling test and the Shapiro–Wilk test. Since all the datasets followed a normal distribution, parametric tests were applied. The data were transformed to fold-change values or relative ratios for analysis where appropriate. For comparisons across multiple groups, a one-way ANOVA with appropriate post hoc multiple comparison tests was used. The statistical significance was set at *p* < 0.05, indicated by asterisks (*) in the figures. A summary of the specific statistical tests applied for each tissue type or experimental condition is provided in Table 1.

## 5. Conclusions

Our findings align with and extend previous work describing magnetic mitohormetic metabolic adaptations. We confirm that weekly PEMF treatment modulates metabolism via systemic adaptations that are largely adipogenic in nature and manifested as increases in adipose mitochondrial expression and thermogenic capacity combined with decreases in cytotoxic and/or mitotoxic ceramide levels. In this study, we implicated the microbiome in these adipogenic adaptations by showing that they could be subsequently observed in HFD-fed recipient mice following FMT from exercised or PEMF-treated mice and were paralleled by improvements in glucose tolerance. Microbiome donor–recipient adipogenic reciprocity was further extended to the liver. Finally, we demonstrated that bone density could be augmented by PEMF treatment and could be reproduced in metabolically compromised recipient mice, characterized by bone loss, by FMT from PEMF-exposed mice. These findings highlight the therapeutic potential of non-invasive PEMF therapy to recapitulate and, in some cases, surpass the capacity of exercise to enhance bone regeneration, promote adipose thermogenesis, improve hepatic lipid metabolism, and instigate positive gut microbiome remodeling. Importantly, all these adaptations, to varying degrees, could be later detected in the recipient mice following FMT from PEMF-exposed mice. However promising, these preclinical indices need to be followed up in human clinical trials. PEMF therapy may potentially provide a simple method to condition the human microbiome for improved bone and metabolic health.

## Figures and Tables

**Figure 1 ijms-26-05450-f001:**
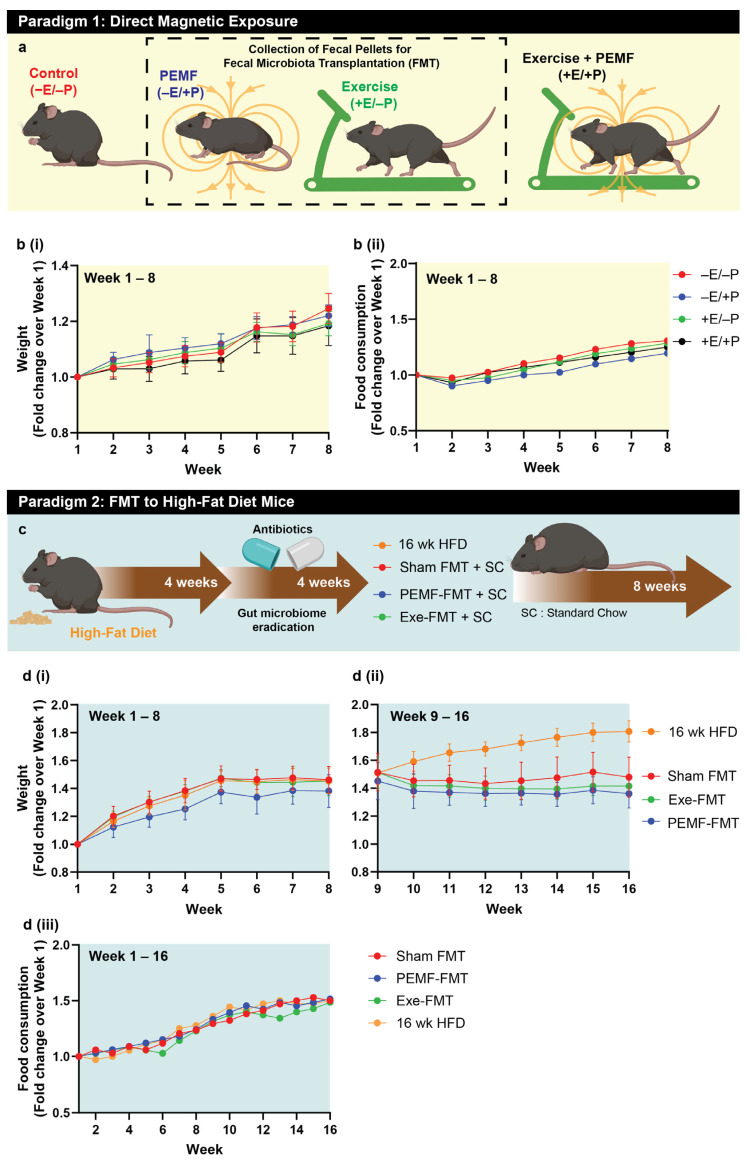
Study design and workflow. (**a**) Paradigm 1 consisted of C57BL/6 mice undertaking 8-week interventions including the following: red—no treatment (−E/−P); blue—PEMF exposure (−E/+P; 1.5 mT for 10 min weekly); green—exercise (+E/−P; twice weekly); or black—combined treatments (+E/+P). Fecal/cecal samples were collected from PEMF and exercise groups for subsequent fecal microbiota transplantation (FMT). Line graphs from the mice of paradigm 1 showing weight gain (**b**(**i**)) and food consumption (**b**(**ii**)) as a fold change from baseline (n = 10 mice per group). (**c**) Paradigm 2 involved recipient mice being fed a high-fat diet (HFD) for 8 weeks (including 4 weeks of antibiotics) before receiving twice-weekly oral gavages of donor microbiota (PEMF-FMT (n = 15) or Exe-FMT (n = 13)) or saline (Sham FMT (n = 12)) for 8 weeks on standard chow. One group of mice was maintained on the HFD during the FMT intervention period (n = 15). (**d**) Body weight changes in recipient and mice maintained on HFD (**i**) before and (**ii**) after commencement of FMT. All weight changes are given as fold change relative to week 1. (**iii**) Line graph shows the food consumption of mice in paradigm 2 over 16 weeks.

**Figure 2 ijms-26-05450-f002:**
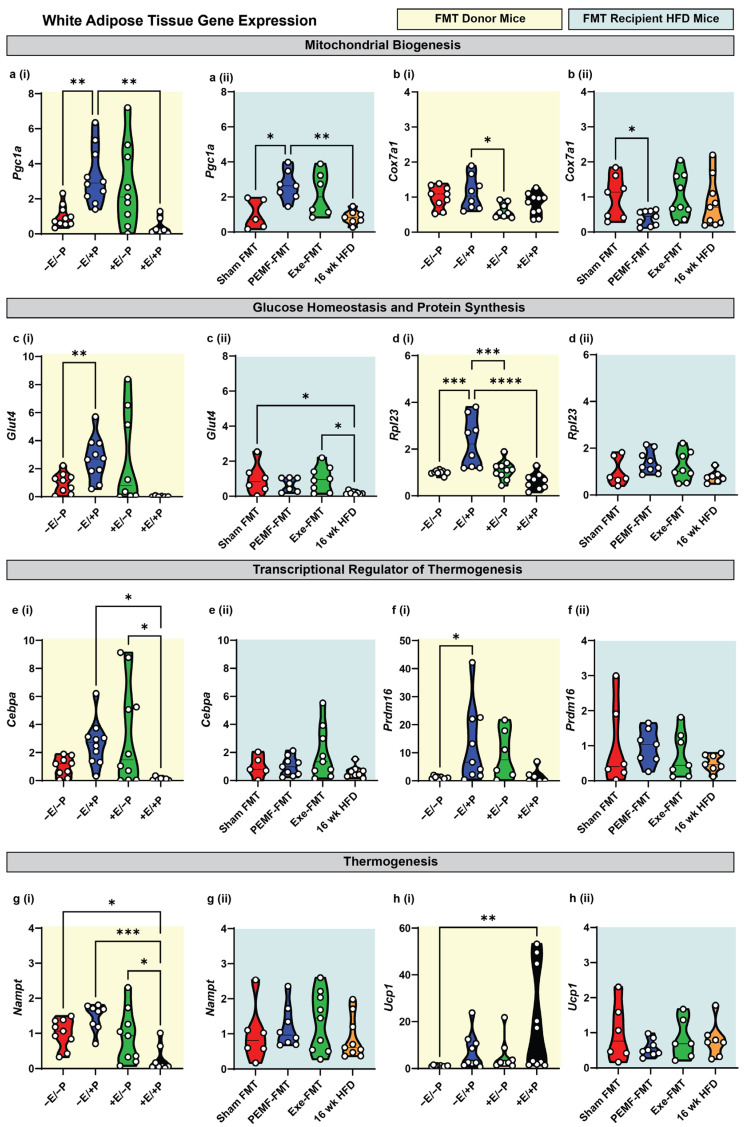
Modulation of metabolic genes in white adipose tissue of donor and FMT recipient HFD mice. Gene expression analysis 2^(−ΔΔCT)^ of mitochondrial biogenesis markers: (**a**) *Pgc1a* and (**b**) *Cox7a1*; glucose homeostasis/protein synthesis markers: (**c**) *Glut4* and (**d**) *Rpl23*; transcriptional regulators of thermogenesis: (**e**) *Cebpa* and (**f**) *Prdm16*; and thermogenesis effectors: (**g**) *Nampt* and (**h**) *Ucp1.* Yellow backgrounds (**i**) indicate FMT donors (paradigm 1; E = exercise; P = PEMF) and light blue backgrounds (**ii**) indicate FMT recipients (paradigm 2; Exe = exercise). Each white dot within the violin plots represents an individual animal. Statistical significance was determined by a one-way ANOVA with Sidak’s post hoc test (* *p* < 0.05, ** *p* < 0.01, *** *p* < 0.001, and **** *p* < 0.0001).

**Figure 4 ijms-26-05450-f004:**
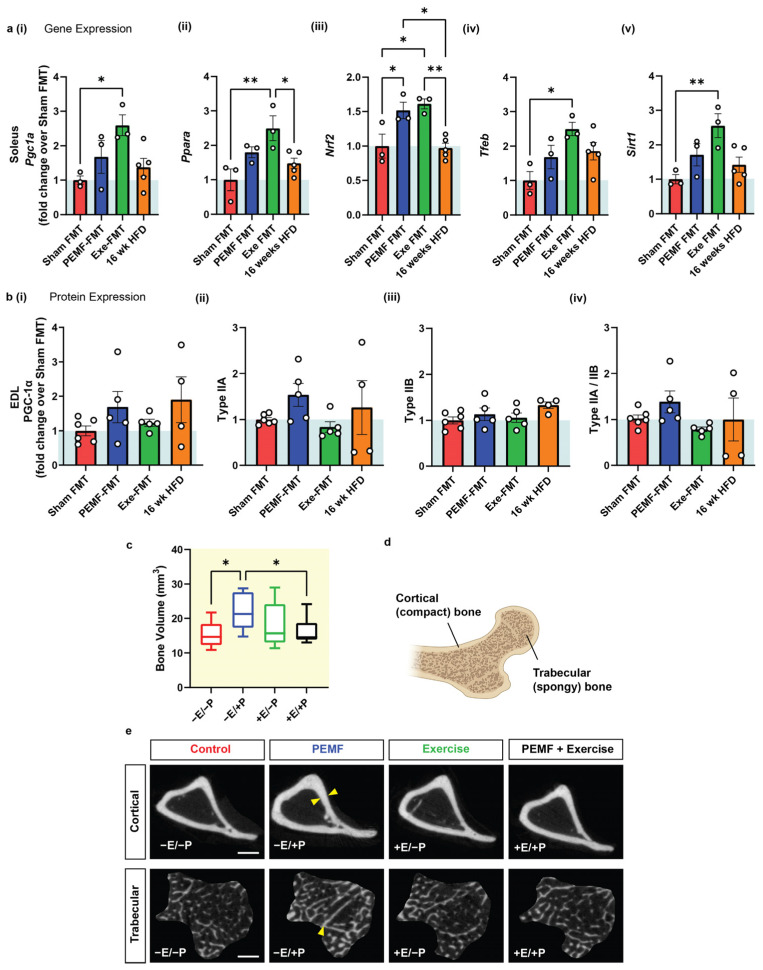
Muscle phenotype analysis and bone volume of FMT donor and recipient HFD mice. (**a**) Bar charts showing soleus 2^(−ΔΔCT)^ gene expression of (**i**) *Pgc1a* (**ii**) *Ppara*, (**iii**) *Nrf2*, (**iv**) *Tfeb*, and (**v**) *Sirt1* in FMT recipients. (**b**) Relative EDL muscle protein analysis of (**i**) PGC-1α, (**ii**) Type IIA, (**iii**) Type IIB, and (**iv**) Type IIA/IIB fiber type expression in FMT recipients. The white dots in the violin plots represent individual data points, with panel (**a**) showing measurements pooled across animals (n = 3–5 mice per group) and panel (**b**) displaying values from individual animals (n = 4–6 mice per group). (**c**) Bone analysis of donor mice showing the bone volume of the tibia (expressed as mm^3^) in control (−E/−P), PEMF-exposed (−E/+P), exercised (+E/−P), and combined treatment (+E/+P) mice (n = 10 mice per group). (**d**) Illustration showing cortical (compact) bone and trabecular (spongy) bone. (**e**) Representative CT images of cortical and trabecular bone under the different treatment conditions. Yellow arrows indicate regions of bone thickening. The quantification of cortical and trabecular bone indices is presented in Figure 5. (E = exercise; P = PEMF). Scale bar = 500 µM. Statistical analysis was performed using a one-way ANOVA and multiple comparisons test. Statistical significance is denoted by * *p* < 0.05 and ** *p* < 0.01, with the error bars representing the standard error of the mean.

**Figure 7 ijms-26-05450-f007:**
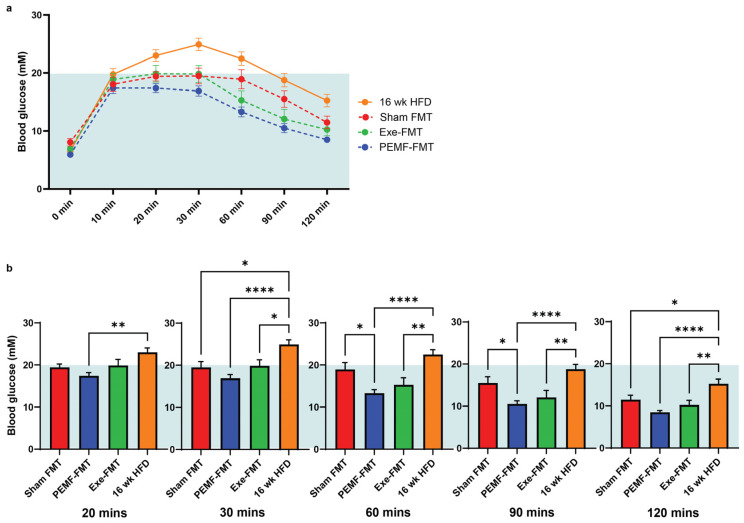
Glucose tolerance analysis of FMT recipient HFD mice. Glucose metabolism was assessed through (**a**) 120 min glucose tolerance curves and (**b**) time-stratified blood glucose levels (mM). The yellow background represents analyses of the FMT donor; the light blue background represents the analyses of the FMT recipient HFD. Data represents the average of n = 12–15 individual mice. Statistical analysis was determined by a one-way ANOVA with Sidak’s multiple comparisons tests (* *p* < 0.05, ** *p* < 0.01, and **** *p* < 0.0001). (E = exercise; P = PEMF).

**Figure 8 ijms-26-05450-f008:**
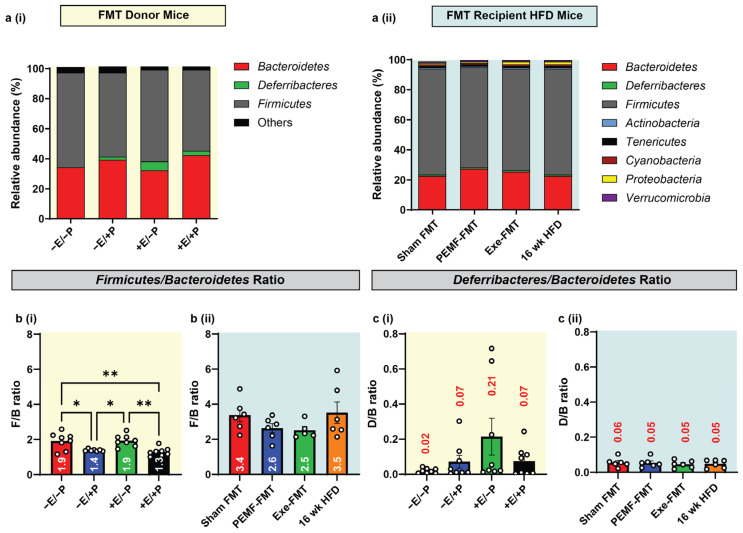
Gut microbiota diversity and composition in donor and recipient mice. (**a**) Phylum-level relative abundance of microbial communities in fecal and cecal samples of (**i**) FMT donor and (**ii**) FMT recipient mice. (**b**) *Firmicutes/Bacteroidetes* (F/B ratio) and (**c**) *Deferribacteres/Bacteroidetes* (D/B ratio) from (**i**) FMT donor and (**ii**) FMT recipient HFD mice. Statistical analysis was performed using a one-way ANOVA with Sidak’s multiple comparisons tests, with * *p* < 0.05 and ** *p* < 0.01. The white dots in the violin plots represent individual animals, with n = 4–8 mice per group. Also see Table 1 (located at the end of the discussion). (E = exercise; P = PEMF).

**Figure 9 ijms-26-05450-f009:**
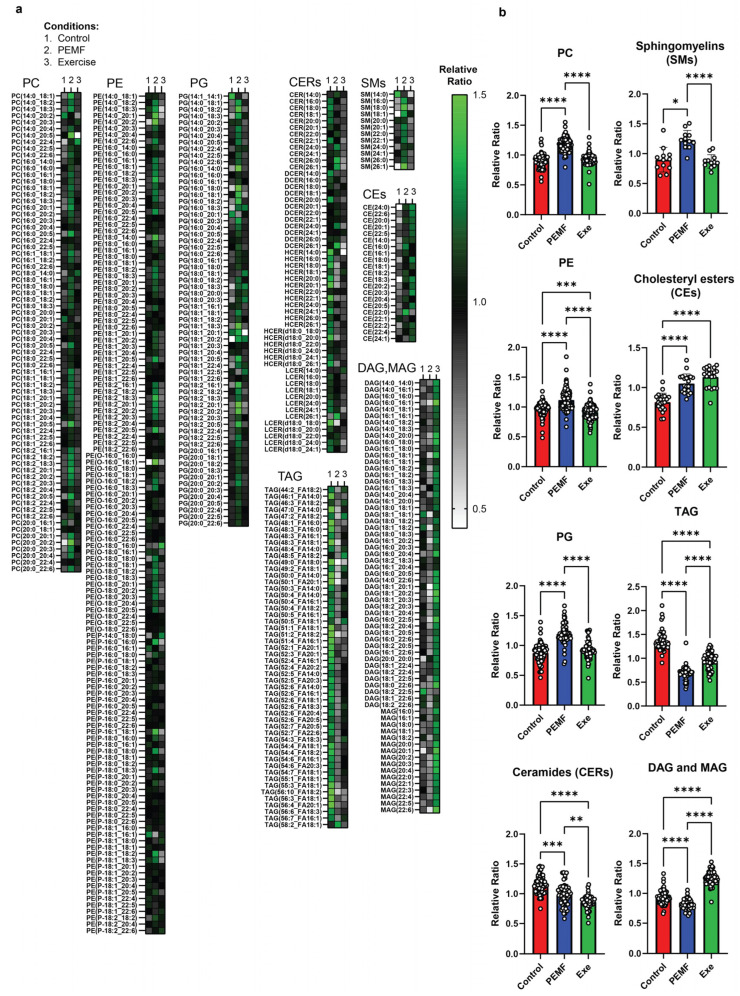
Hepatic lipid profiles from donor mice. LC-GC/MS analyses of individual liver lipid subspecies within a lipid class expressed as a relative ratio (n = 5 mice per group). (**a**) Heatmap depiction of phospholipids (PC, PE, and PG), sphingolipids (CERs and SMs), cholesteryl esters (CEs), and neutral lipids (TAG, DAG, and MAG). (**b**) Bar charts display the mean relative ratio of all lipid subspecies combined, while white dots represent the average relative ratios of individual lipid subspecies from 5 animals per treatment group. Statistical analysis was determined using a one-way ANOVA with Tukey multiple comparisons test. Significant differences between treatment groups are denoted as * *p* < 0.05, ** *p* < 0.01, *** *p* < 0.001, and **** *p* < 0.0001.

**Figure 10 ijms-26-05450-f010:**
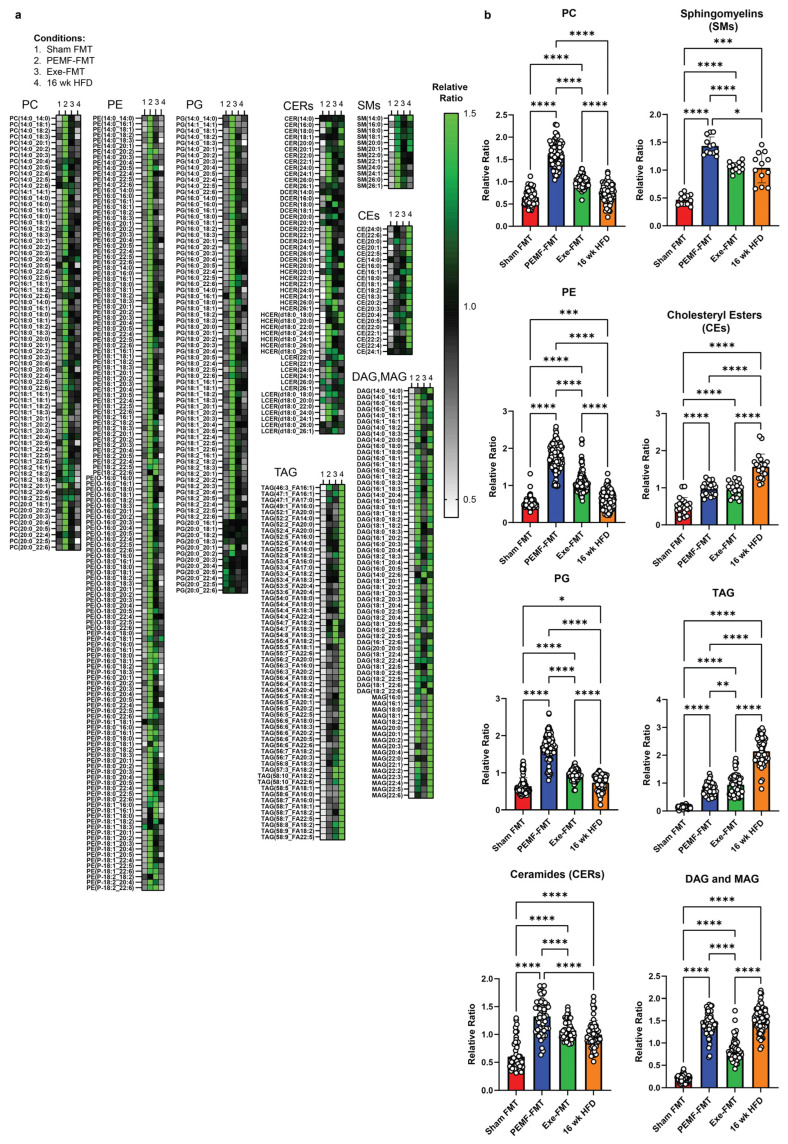
Hepatic lipid profiles from HFD recipient mice. LC-GC/MS analysis of liver lipids with each subspecies within a lipid class expressed as a relative ratio (n = 4–6 mice per group). (**a**) Heatmap visualization of phospholipids (PC, PE, and PG), sphingolipids (CERs and SMs), cholesteryl esters (CEs), and neutral lipids (TAG, DAG, and MAG). (**b**) Bar charts display the mean relative ratio of all lipid subspecies combined, while white dots represent the average relative ratios of individual lipid subspecies from 4-6 animals per treatment group. Statistical analysis was performed by comparing the mean of relative ratios using a one-way ANOVA with Tukey multiple comparisons test. Significant differences between treatment groups are denoted as * *p* < 0.05, ** *p* < 0.01, *** *p* < 0.001, and **** *p* < 0.0001.

**Table 2 ijms-26-05450-t002:** List of mouse qPCR primers in adipose tissue samples.

Gene	Forward Primer Sequence (5′…3′)	Reverse Primer Sequence (5′…3′)
*AdipoQ*	GCACTGGCAAGTTCTACTGCAA	GTAGGTGAAGAGAACGGCCTTGT
*Leptin*	GAGACCCCTGTGTCGGTTC	CTGCGTGTGTGAAATGTCATTG
*Cox7a1*	CAGCGTCATGGTCAGTCTGT	AGAAAACCGTGTGGCAGAGA
*Cebpa*	TTCGGGTCGCTGGATCTCTA	TCAAGGAGAAACCACCACGG
*Glut4*	ACGTTGGTCTCGGTGCTCTT	GGCCACGATGGAGACATAGC
*Nampt*	CATTCAAGGAGATGGCGTGG	CCTTAAACACATTAACCCCAAGGC
*Prdm16*	AGTCCTCCATACCAGGAGCTG	CCAAGTCTTCAGAGATCTGCTTTT
*Rpl23*	AGATGTCGAAGCGAGGACGC	GTCTGTTCAGCCGTCCCTTG
*Ucp1*	ACTGCCACACCTCCAGTCATT	CTTTGCCTCACTCAGGATTGG
*Pgc1a*	GGAGTGACATAGAGTGTGCTG	TGGTCGCTACACCACTTCAA
*Paqr4*	CAGCCTTTTCTACCTACACAACG	GCACATGAAGAGGTGATACAGCA
*B2m*	GATGTCAGATATGTCCTTCAGCA	TCACATGTCTCGATCCCAGT

**Table 3 ijms-26-05450-t003:** List of mouse qPCR primers in muscle tissue samples.

Gene	Forward Primer Sequence (5′…3′)	Reverse Primer Sequence (5′…3′)
*Pgc1a*	GGAGTGACATAGAGTGTGCTG	TGGTCGCTACACCACTTCAA
*Sirt1*	TGACCGATGGACTCCTCACT	ACAAAAGTATATGGACCTATCCGC
*Nrf2*	TGAAGCTCAGCTCGCATTGA	TGCTCCAGCTCGACAATGTT
*Tfeb*	TGTCTAGCAGCCACCTGAAC	GCTCTGCTCTCAGCATCTGT
*Ppara*	GCAACCATCCAGATGACACC	TCTCTTGCAACAGTGGGTGC
*B2m*	GATGTCAGATATGTCCTTCAGCA	TCACATGTCTCGATCCCAGT

## Data Availability

All the data generated in this study are presented in this article. The raw data supporting the findings are available from the corresponding author upon request.

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
