# Peer review of "Fecal Microbiota Transplantation from Mice Receiving Magnetic Mitohormesis Treatment Reverses High-Fat Diet-Induced Metabolic and Osteogenic Dysfunction"

_ijms, 2025, doi:10.3390/ijms26125450_

Round 1
Reviewer 1 Report
Comments and Suggestions for Authors
The article Fecal Microbiota Transplantation from Mice Receiving Magnetic Mitohormesis Treatment Reverses High-Fat Diet-Induced Metabolic and Osteogenic Dysfunction presents data related to fecal microbiota transfer. Recommendations:
- The Introduction should be rewritten. It is much too long and includes mixed content, combining elements from the Discussion and Materials and Methods sections.
- Figure 1 should be relocated.
- The number of citations in the first section is excessive and not typical for a scientific article.
- The results are interesting, but the statistical analysis is rather weak. More complex analyses should be added.
- The sample size is quite small. What is the power of the study?
- Clearly define the control group.
- In the Discussion section, you could include a table summarizing the key articles in the literature. I also recommend extrapolating the findings to humans and discussing the role of fecal microbiota transplantation in metabolic syndrome – suggested reference: 10.3390/jcm14082678.
- The Conclusions section should not contain citations and must clearly summarize the key findings of the article.
Reviewer 2 Report
Comments and Suggestions for Authors
Using a mouse fecal microbiota transplantation model, the authors examined the consequences of introducing fecal microbiota from mice that have been treated with pulsed electromagnetic field therapy and/or exercise. The authors reported less body weight gain, improved insulin sensitivity, and enhanced bone density with PEMF-FMT. They showed that PEMF treatment enhanced PGC-1α-associated mitochondrial, metabolic, and thermogenic gene expression. These beneficial effect was also observed in the recipient mice when fecal matter transplant is performed.
There are a few major concerns about the paper:
- With fecal matter transplant, materials in the fecal including probiotics and/or prebiotics are introduced to the recipients. The reason that phenotype/metabolic beneficial effect seen with the donor was also present in the recipients is more than likely from the probiotics and/or prebiotics present in the fecal matter of the donor. It should not be described as the phenotype (ie. the oxidative capacity at line 55) of the recipients being transferred from the donor.
Oxidative capacity, a reflection of the muscle's ability to utilize oxygen for energy production during physical activity, is a specific characteristic of an individual's muscle tissue, determined by factors like training, genetics, and health conditions, not something that can be transferred.
- The fecal matter transplant is not well described. There is no description as to how the fecal matter was processed after collection? Was the whole fecal matter collected used as is for the transplant? how much of the fecal matter was used for each gavage? Was it controlled by weight? By the amount of microorganism present? These information are all missing in the paper.
- There is no food consumption data for the animals on each treatment. Are they consuming the same amount of food? Do they have the same amount of fecal matter /per day?
- Line 665: what is the interval for “fresh fecal pellets” collection? Every 2 hr? 8 hrs? 24 hrs? How frequent were the fecal collected from the cage? What is the total amount of fecal matter released/24hr?
- What is the body temperature of the mice in each group? The animals have decreased body weight gain with PEMF-FMT. There are increased enhanced PGC-1α-associated mitochondrial, metabolic, and thermogenic gene expression. Does it translate into increased basal body temperature of the mice in this group?
- Please check the instruction to Authors, and the format for listing of vendors :
Line 802: Agilent 6430 triple- 802 quadrupole mass spectrometer (XXX)
Line 826: using LICOR Odyssey FC imaging system (XXX). 826
Line 842: Using Galaxy (XXX). 842
Line 844: Statistical analysis and data were generated using GraphPad Prism 10 (XXX).
- “For the first time we demonstrate that bone density is augmented by PEMF treatment and could be likewise transmitted via FMT to metabolically compromised recipient mice.”
The concept needs to be clarified here again.
Would the authors mean that: the effect to bone density by PEMF can also been seen in the recipient mice. Please note that bone density also cannot be transferred.
- Figure 3, please show scale bar on each of the picture. Please make sure all of the pictures are on the same scale.
The english should benefit from the help of a professional language editor.
Round 2
Reviewer 1 Report
Comments and Suggestions for Authors
The authors have made the requested changes.
Author Response
We thank the reviewer for taking time to read our manuscript. The input has been valuable to strengthen this manuscript.
Reviewer 2 Report
Comments and Suggestions for Authors
The authors presented very good improvement of the manuscript with the revision.
Some minor concerns:
Line 193: whereas FMT from exercised mice did not.
Please be consistent. This treatment was labeled as “Sham-FMT” in the figure.
Line 316: Blood sugar tolerance
Blood glucose tolerance
Author Response
We sincerely thank the reviewer for the prompt evaluation of our manuscript and for their second feedback. We apologize for the oversight regarding the label standardization (Sham FMT, Exe-FMT etc) and terminology (sugar to glucose) and have carefully addressed them in the revised manuscript, with tracked changes for clarity.